# Antimicrobial Peptides Mediate Apoptosis by Changing Mitochondrial Membrane Permeability

**DOI:** 10.3390/ijms232112732

**Published:** 2022-10-22

**Authors:** Hongji Wang, Chaowen Zhang, Mengnan Li, Chaoran Liu, Jingyi Wang, Xuan Ou, Yuzhu Han

**Affiliations:** 1College of Animal Science and Technology, Southwest University, Chongqing 402460, China; 2Immunology Research Center, Medical Research Institute, Southwest University, Chongqing 402460, China

**Keywords:** AMPs, mitochondrial outer membrane permeability (MOMP), mitochondrial inner membrane permeability (MIMP), mitochondrial permeability transition (MPT), apoptosis

## Abstract

Changes in mitochondrial membrane permeability are closely associated with mitochondria-mediated apoptosis. Antimicrobial peptides (AMPs), which have been found to enter cells to exert physiological effects, cause damage to the mitochondria. This paper reviews the molecular mechanisms of AMP-mediated apoptosis by changing the permeability of the mitochondrial membrane through three pathways: the outer mitochondrial membrane (OMM), inner mitochondrial membrane (IMM), and mitochondrial permeability transition pore (MPTP). The roles of AMPs in inducing changes in membrane permeability and apoptosis are also discussed. Combined with recent research results, the possible application prospects of AMPs are proposed to provide a theoretical reference for the development of AMPs as therapeutic agents for human diseases.

## 1. Introduction

In recent decades, the incidence and mortality of malignant tumors have remained high, and there are difficulties, such as low treatment efficiency and drug resistance [1]. In addition, the emergence of resistant pathogenic microorganisms might cause a variety of infectious diseases [2], which seriously threaten human health. Antimicrobial peptides (AMPs) have good inhibitory effects on bacteria, fungi, viruses, and even tumor cells, without altering drug resistance [3,4]. This advantage provides a prerequisite for the development of AMPs as therapeutic agents. Previous studies have shown that AMPs can not only directly act on cell membranes and cell walls to kill cells, but can also enter cells and target organelles to achieve bactericidal effects by interfering with normal metabolic processes and playing an immunomodulatory role [5,6,7]. Many AMP drugs have been developed for clinical treatment, such as lipopeptide A21978C for the treatment of skin infection and sepsis and peptide Glutoxim/NOV-002 for the treatment of tuberculosis and non-small-cell lung cancer [8]. To address the resistance of antibiotics, the design and production of effective AMP preparations has become inevitable. However, with changes in the structure, hydrophobicity, and amphipathic properties of AMPs, the mechanism and target of action would also change [5,9]. The therapeutic mechanism of AMPs that exerts antitumor or antifungal effects through the cell membrane or wall is well-understood [10,11]. Nevertheless, studies of AMPs that cause apoptosis through the intracellular pathway are still insufficient. Therefore, sorting out the mechanism of AMPs for the treatment of tumors and infectious diseases is convenient for screening AMPs for better efficiency and to develop them as therapeutic drugs.

Through the investigation of the mechanism of action of different AMPs, it was found that some AMPs cause damage to mitochondria when exercising their biological functions [12,13,14]. Mitochondria is a site for energy production in cells, playing an important role in regulating cell growth and death [15]. The integrity of the outer and inner mitochondrial membrane is crucial for mitochondrial apoptosis, and even cell apoptosis. Changes in mitochondrial membrane permeability can directly activate the mitochondrial apoptosis pathway [16,17]. Although there are abundant reports on the effects of AMPs on mitochondria, most of the research results are merely descriptions of the phenomena, such as mitochondrial membrane potential depolarization and oxidative stress, causing mitochondrial dysfunction [18]. Only few reports have studied the effect of AMPs on mitochondrial permeability transition pore (MPTP) in detail [19], and there are no detailed classifications and mechanism analyses of AMPs on mitochondrial membrane damage. In fact, in the process of the endogenous pathway of apoptosis (the mitochondrial pathway), both the outer and inner membranes of mitochondria are damaged to varying degrees [20]. Combined with the current research reports, the damage to mitochondrial integrity is simply divided into three categories: mitochondrial outer membrane permeability (MOMP), mitochondrial inner membrane permeability (MIMP), and mitochondrial permeability transition (MPT). Separately discussing the effects of AMPs on mitochondrial membranes, especially those targeted at mitochondria [21], can provide a clearer understanding of their therapeutic mechanisms and facilitate the study of the mechanisms of unknown AMPs.

This paper systematically discusses the ways in which AMPs alter mitochondrial membrane permeability. The effect of AMPs during apoptosis will be analyzed from the perspective of MOMP, MIMP, and MPT. Some structural features of AMPs targeting the mitochondrial outer and inner membrane are also discussed, which may provide new ideas for the artificial modification of AMPs.

## 2. AMPs and Apoptosis

As a regulated and active programmed death process, apoptosis is very important for the body to maintain homeostasis [22]. There are three main mechanisms regulating apoptosis: the mitochondrial pathway, the death receptor pathway, and the endoplasmic reticulum pathway [14,23]. AMPs are a class of small molecular peptides with no more than 100 amino acids [24]. Most AMPs exert their antimicrobial activity by disrupting cell membranes [25], and some AMPs with cell-penetrating ability can exert antitumor or antifungal activities in cells by different mechanisms. For example, they can bind nucleic acids and inhibit DNA or RNA synthesis to kill bacteria [26]. Alternatively, they can inhibit enzyme activity in the process of nucleic acid and protein anabolic metabolism [7]. In fact, we observed that AMPs that enter the cell or exert their effects are often able to achieve programmed cell death by mitochondrial swelling, outer mitochondrial membrane (OMM) rupture, and by stimulating the activation of apoptotic markers (Table 1). As mentioned above, mitochondrial membrane damage is inevitable during the mitochondrial apoptotic pathway. Therefore, it is necessary to describe the mechanism of AMP-induced apoptosis from the perspective of mitochondrial membrane damage.

## 3. AMPs and MOMP

MOMP refers to the process by which proteins existing in the intermembrane space (IMS) (cytochrome *c* and apoptosis-inducing factor) enter the cytoplasm through the OMM, activate the apoptotic pathway, or trigger inflammatory reactions, thereby causing mitochondrial and cell apoptosis [45,46,47]. MOMP, which does not involve the inner mitochondrial membrane (IMM) and MPTP, is mainly regulated by members of the Bcl-2 family proteins, which contain one or more BH domains. Activated proapoptotic proteins Bak and Bax oligomerize and form pores in the OMM, thus initiating MOMP. Bcl-2 and Bcl-XL are anti-apoptotic proteins that inhibit the activities of Bak and Bax [48,49,50]. PETK, which is a Bax-derived peptide that contains the same BH domain as Bax, can change the permeability of the OMM by the same oligomerization mechanism and subsequently activate the apoptosis program mediated by caspase-3 [33]. This kind of artificially modified AMPs, such as PETK targeting the OMM, are good candidate drugs for the treatment of tumors and infectious diseases [21]. LL-37 accumulates in the mitochondria of human osteogenic MG63 cells and promotes the release of cytochrome *c* (Cyt *c*) and apoptosis-inducing factor (AIF) from IMS into the cytoplasm. Meanwhile, we could detect leaked Cyt *c* and AIF in the cytoplasm, while cytochrome *c* oxidase IV (COXIV), located in the IMM, was not detected [30]. This finding indicates that LL-37 may only activate the permeability of the OMM, without damaging the IMM structure [30]. However, the specific mechanism of the change in the permeability of the mitochondrial outer membrane remains to be explored.

As a derivative peptide of LL-37, 17BIPHE2 can regulate the expression of Bax and Bcl-2 in cancer cells by activating the ERK pathway to induce cell apoptosis [27]. Notably, the mitochondrial crest structure was damaged during this process, which may be attributed to the increase in intracellular reactive oxygen species (ROS) and Ca^2+^ concentration caused by 17BIPHE2. It is possible that MOMP, MPTP, and MIMP occur simultaneously, and the three processes act together to disrupt the structure of mitochondria. Humanin (HN) is an effective mitochondrial-derived peptide that inhibits apoptosis. It has been reported that HN is able to induce structural changes in Bax, interact with structurally altered Bax, and produce fibers, thereby sequestrating Bax and preventing the initiation of MOMP [28,29]. Besides, some studies found that, during MOMP, factors such as Cyt *c*, Smac, and AIF are first released into the cytoplasm. After a series of reactions, Cyt *c* and Apaf-1 form apoptotic bodies, which activate the caspase cascade and lead to apoptosis [51]. At the same time, Smac can prevent the anti-apoptotic effect of XIAP. If caspase is absent or inactivated during MOMP, several apoptotic factors can induce apoptosis via inflammation (Figure 1) [52].

According to the mechanisms of action of PETK, LL-37, 17BIPHE2, and HN [27,28,30,33], we can conclude that AMPs initiate MOMP through two distinct pathways, leading to apoptosis. One way is the direct participation of AMPs in MOMP to promote cell apoptosis with the same specific framework of apoptotic proteins regulating MOMP. These AMPs are usually artificially modified proapoptotic proteins [33]. The other way is to regulate the expression of apoptotic proteins, in order to indirectly regulate MOMP and subsequent apoptosis [27,31]. In the absence of caspases, AMPs can also trigger inflammation with the release of pro-apoptotic factors. Although increased expression of activated caspase-3 in the apoptotic pathway was detected [33,43], it is not clear whether this is caused by the apoptotic chain reaction (the combination of Cyt *c* with Apaf-1 to trigger apoptosis) induced directly by AMPs or by the AMP-induced expression of caspase genes. Because of the composition and structural characteristics of AMPs and the mechanism of their multi-target action, the simultaneous existence of these two causes is not excluded, but this still needs to be supported by more thorough evidence. For now, we can draw a conclusion that most AMPs promote apoptosis indirectly by promoting MOMP.

## 4. AMPs and MIMP

MIMP is considered to be a process that occurs after MOMP, during which substances in the mitochondrial matrix are released into the cytoplasm across the IMS. Theoretically, when MOMP is initiated, as Bak/Bax continues to oligomerize in the OMM, the forming pores will also continue to expand. The IMM is squeezed and protrudes from the pores on the OMM, which eventually leads to an increase in the permeability of the IMM and the release of mitochondrial DNA (mtDNA) [53,54]. Of course, if the degree of MIMP is mild, some of the ions in the mitochondrial matrix are allowed to reach the cytoplasm, causing changes in mitochondrial membrane potential. Under pathological conditions, mtDNA can activate the cGAS-STING signaling pathway in the cytoplasm, the signal of which is transmitted to the nucleus to generate interferon β (IFN-β) and trigger an inflammatory response (Figure 1) [55,56]. Moreover, it has been reported that mtDNA release is independent of mitochondrial fission and MPT [57]. Therefore, the leakage of mtDNA is one of the markers to determine the occurrence of MIMP. In fact, apoptotic caspases can inhibit inflammatory reactions during apoptosis [58], which explains why it is difficult to observe inflammatory reactions during MOMP, even if the apoptotic factors involved in inflammation are released.

Some Arg-rich artificial peptides, such as KLAKLAK2 (KLA), can cause a decrease in mitochondrial membrane potential, but without initiating MOMP. Moreover, the ability of KLA to penetrate the outer and inner mitochondrial membrane is dependent on the mitochondrial membrane potential [35]. Under normal circumstances, any impairment of mitochondrial function will affect the mitochondrial membrane potential. Thus, the decrease of mitochondrial membrane potential is considered to be a preceding event to apoptosis [59,60]. Early research on cecropin A–melittin, a short hybrid peptide, found that it could pass through the IMM and allow the entry and exit of some factors, but it would destroy the structure of the IMM [34]. In a study of mitochondria-mediated apoptosis, when cells were treated with the BH3-mimetic ABT-737, which can act as an inhibitor of Bcl-XL and Bcl-2, we observed that MOMP was followed by mtDNA leakage [57]. However, it is not clear whether AMPs have a direct effect on MIMP. How mtDNA can cross the IMM and reach the cytoplasm without damaging the IMM structure remains our focus of attention. According to the premise that AMPs induce MOMP, it is speculated that AMPs can also induce MIMP and amplify its permeability. This idea could help explain why mtDNA leaks. Or perhaps, contrary to our suspicion, the leakage of mtDNA is accompanied by structural disruption of the IMM. After all, there are huge temporal differences between MOMP and MIMP [57], whereas severe MOMP-induced mitochondrial dysfunction is sufficient to disrupt mitochondrial structure. In either case, we need to continue to explore the specific mechanisms of MIMP.

## 5. AMPs and MPTP

MPTP contains three important components: the voltage-dependent anion channel (VDAC) located in the OMM, adenine nucleotide transporter (ANT) located in the IMM, and cyclophilin (Cyp-D) located in the mitochondrial matrix (Figure 2). These three components are essential for maintaining mitochondrial osmotic balance and mitochondrial membrane permeability [61,62,63]. Under normal physiological conditions, MPTP is in a transient open state. If it is in a continuous open state, MPT occurs, which can lead to the extensive swelling of the mitochondrial matrix, accompanied by the rupture of the OMM, collapse of mitochondrial membrane potential, depletion of cell ATP, and, ultimately, cell necrosis or apoptosis [64]. Dithiothreitol (DTT) can inhibit MPTP caused by sulfhydryl oxidation of the OMM proteins, while cyclosporin A (CsA) can interact with Cyp-D, both of which are typical MPTP opening inhibitors (Figure 2) [65]. CGA-N12, a derivative peptide of CGA-N46, can eliminate their inhibitory effect and improve mitochondrial permeability [38]. In addition, CGA-N12 maintains the MPTP opening by inducing the accumulation of intracellular ROS, thus leading to cell apoptosis [65]. Ca^2+^ overload in the mitochondrial matrix is also an important factor contributing to the sustained opening of the MPTP. BIRD-2 is a synthetic peptide that interferes with the inositol 1,4,5-triphosphate receptor (IP3R), which independently induces apoptosis and appears to promote mitochondrial Ca^2+^ uptake, followed by sustained opening of the MPTP [36,66].

Compared with mitochondrial membrane damage involving only MOMP, MPT induces relatively severe damage and other changes. The continuous opening of the MPTP will also directly cause MOMP and MIMP, and even damage the structure of mitochondria. In addition, there are many factors affecting the opening of the MPTP, and mitochondrial damages are also the result of a combination of factors. On the one hand, tilapia antimicrobial peptide 3 (TP3) can induce an increase in mitochondrial and intracellular ROS, thus promoting the opening of the MPTP. On the other hand, TP3 affects mitochondrial fission, leads to mitochondrial dysfunction, and increases the probability of MPTP opening [43]. Finally, TP3 leads to enhanced MPT, thereby prompting the activation of caspase-3/9 [39], which greatly facilitates apoptosis. *Litopenaeus vannamei* hemocyanin-derived peptide can also change mitochondrial permeability and mediate apoptosis by reducing the mitochondrial membrane potential and increasing the expression of caspase-3/9 and Bax [40].

All the above reports indicate that AMPs could induce continuous MPTP opening and promote apoptosis. The greater the effect of AMPs on mitochondrial permeability, the stronger their cytotoxicity [37]. After treatment with AMPs, Cyt *c* is released, ROS is increased, and the mitochondrial membrane potential is disrupted [41]. Altogether, these results confirm that MPT induced by AMPs exhibits the typical characteristics of MOMP and MIMP, besides inducing apoptosis in the same manner. In addition, AMPs strengthened the necessary link between MPTP and MOMP (Figure 1). We can infer that MIMP occurs during MPT. It is not clear whether MIMP is directly caused by the continuous opening of MPTP or by MOMP.

## 6. Challenges for Antimicrobial Peptides

Drugs that exploit the death receptor pathway to trigger the death of tumor cells and antibiotics used to treat infectious diseases can cause cells to become resistant [5,67]. Mitochondria are key to the growth and metastasis of tumor cells and play an important role in the growth inhibition of pathogens [68]. Although AMPs can replace antibiotics to exert therapeutic effects [69], the disadvantages of AMP application are very obvious. AMPs extracted from natural biological resources have low content and high extraction cost. Therefore, synthetic peptides may become a new breakthrough. By using various modifications, such as acetylation, glycosylation, protein fusion, and cyclization, the structure of AMPs can be optimized, the difficulty of industrial synthesis can be reduced, and the therapeutic effect can be improved [70,71]. For AMPs that damage the mitochondrial membrane, but cannot penetrate it, we can choose appropriate methods to render them membrane permeant. For example, AMPs modified by hydrophilic arginine–glycine–aspartate (RGD) sequence by protein fusion have the ability to penetrate cell membranes. This modification allows for the transport of AMPs from the extracellular to the intracellular space [72]. Therefore, it is feasible to rationally design AMPs to render them membrane permeant or to enhance their binding to mitochondrial targets [73], which is of great significance for improving their therapeutic effect.

In addition, almost all AMPs that can change mitochondrial membrane permeability have α-helices or can form ring structures, and their targets may be related to the structure and activity of AMPs [74] (Table 1). AMPs are difficult to obtain in large quantities, due to their low content in living organisms. Moreover, some AMPs have problems of cytotoxicity and bioavailability in clinical application [75]. Because the cytotoxicity of an AMP is positively correlated with its effect on mitochondrial outer and inner membrane permeability [37], we need to select an AMP that can alter mitochondrial membrane permeability without severely damaging mitochondrial structure and without affecting other organelles. We can, therefore, select less cytotoxic AMPs, according to the degree of mitochondrial structure damage. One of the known mechanisms of AMP membrane targeting is the cationic interaction of AMPs with negatively-charged cell membranes to increase their permeability [76]. Both the cell membrane and OMM have similar components (phospholipids and proteins) and electric charge, so it is not clear whether similar interaction of AMPs with the cell membrane can also occur on the OMM. Only after we understand the mechanism of action of AMPs can we modify them into therapeutic drugs. In addition, increasing the yield of AMPs and optimizing the technology of the modification of AMPs are among the challenges that we are currently facing.

## 7. Conclusions

We conclude that AMPs are potent inducers of apoptosis. If the AMPs are stable and persistent to induce MOMP, the pores formed on the OMM will persist for a long time and show a trend of continuous expansion until the AMPs are inactive, so MIMP is inevitable. On the one hand, the change in mitochondrial membrane potential will cause the abnormal continuous opening of the MPTP [38]. On the other hand, AMPs affect the opening of the MPTP by regulating the production of ROS, the concentration of Ca^2+^, or targeting the MPTP components [44]. Because MPTP spans the outer and inner membranes of mitochondria, it must also affect MOMP and MIMP [41]. In summary, the three pathways of membrane permeability changes induced by AMPs are interconnected, and even mutually reinforced (Figure 3). The mechanism of apoptosis caused by these three pathways has been confirmed by many experiments: apoptosis is mediated by the caspase protein cascade, triggered by the release of Cyt *c* [77] and the caspase-independent pathway, which is mediated by inflammatory signals, such as AIF or mtDNA [78,79].

After entering cells, AMPs directly or indirectly change the permeability of the mitochondrial outer and inner membranes, thereby activating the mitochondrial pathway [32]. Irrespectively of whether AMPs affect MOMP, MIMP, or MPT first, these three changes may eventually occur simultaneously in mitochondria. This may be one of the reasons why some AMPs have strong therapeutic effects and occasionally lead to the disruption of the mitochondrial structure. Therefore, AMPs that mediate mitochondrial apoptosis often simultaneously change the permeability of the outer and inner mitochondrial membrane in a synergistic manner [42]. In this review, the possible mode of action of AMPs in changing mitochondrial permeability is described, and the relationship between AMPs and MOMP, MIMP, and MPT is analyzed. This can facilitate the investigation of the mechanism of action of unknown AMPs and the screening and development of known AMPs. Undeniably, there are many unrevealed relationships regarding the effect of AMPs on apoptosis, such as whether the increase in caspase-3 expression is directly related to AMPs and whether AMPs are directly related to MIMP. All of these questions deserve further research and exploration.

## Figures and Tables

**Figure 1 ijms-23-12732-f001:**
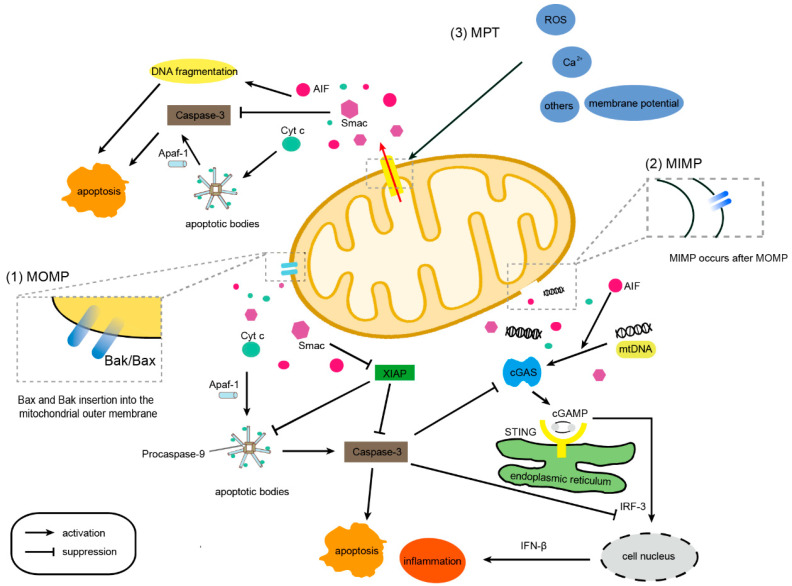
Main apoptosis mechanisms induced by MOMP, MIMP, and MPT. (1) Bak/Bax aggregates and forms pores in the OMM. Cyt *c*, AIF, Smac, and other molecules leak from the pores. After a series of reactions, Cyt *c*, Apaf-1, and procaspase-9 assemble into apoptotic bodies and subsequently activate caspase-3. Caspase-3 inhibits inflammation and induces apoptosis. The released Smac can inhibit the antiapoptotic effect of XIAP. (2) MIMP occurs after MOMP. Under pathological conditions or in the absence of caspases, the cGAS-STING signaling pathway can be activated by mitochondrial DNA (mtDNA) and AIF to release IFN-β and other inflammatory factors, thus triggering inflammation. (3) Changes in intracellular ROS, Ca^2+^ concentration, or mitochondrial membrane potential depolarization can induce MPTP to remain open, leading to changes in both outer and inner mitochondrial membrane permeability. During MPT, Cyt *c*, AIF, Smac, and other molecules are released, inducing apoptosis in a caspase-3-dependent manner.

**Figure 2 ijms-23-12732-f002:**
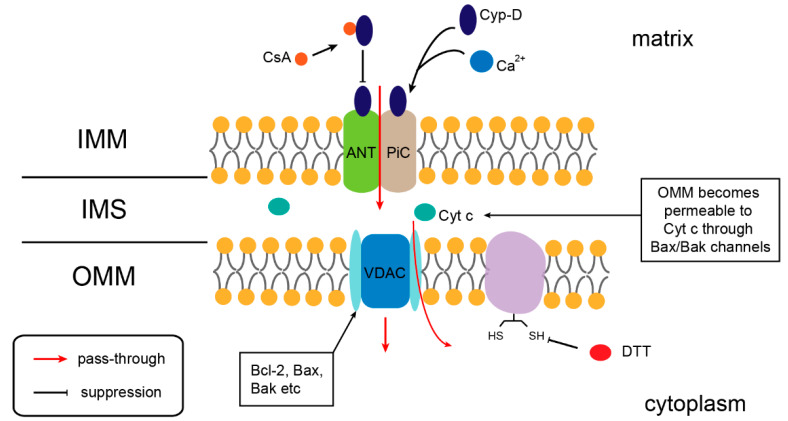
The structure of MPTP. Adapted with permission from Ref. [64]. 2015, Halestrap, A.P. IMM: inner mitochondrial membrane; OMM: outer mitochondrial membrane; IMS: intermembrane space; Cyt *c*: cytochrome *c*. Voltage-dependent anion channel (VDAC), adenine nucleotide transporter (ANT), cyclophilin (Cyp-D), and the phosphate carrier (PiC) are components of the MPTP. Cyclosporine A (CsA) can interact with Cyp-D and inhibit MPTP opening. Dithiothreitol (DTT) inhibits the MPTP opening caused by sulfhydryl oxidation of OMM proteins.

**Figure 3 ijms-23-12732-f003:**
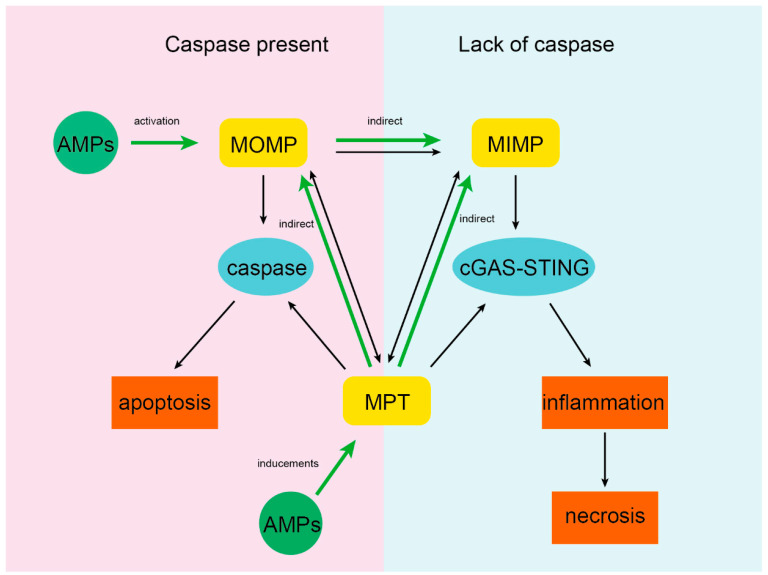
Effects of AMPs on MOMP, MIMP, and MPT. On the one hand, AMPs can induce MOMP and MPT directly. On the other hand, because MOMP, MIMP, and MPT are closely related, AMPs can indirectly induce MIMP. AMPs can trigger inflammation or apoptosis in cells by affecting MOMP, MIMP, or MPT and then indirectly affecting one or both processes.

**Table 1 ijms-23-12732-t001:** Effect of some AMPs on the outer and inner mitochondrial membrane.

Membrane Targets	Peptide	Source	Structure	Object	Mechanism	References
Outer mitochondrial membrane (OMM)	17BIPHE2	LL-37-derived peptide	α-helices	Lung adenocarcinoma A549 Cells	MOMP was changed by upregulating the expression of Bax and downregulating the expression of Bcl-2	[27]
	Humanin	Mitochondrial-derived peptides	α-helices; a loop structure	In vitro experiment	Combined with Bak and prevented MOMP	[28,29]
	LL-37	White blood cells; epithelial cells	α-helices	Human osteoblast-like MG63 cells	Released Cyt *c* and promoted MOMP	[30]
	LTX-315	Lactoferricin derivative	α-helices	U2OS cells	Released Cyt *c* and promoted MOMP	[31]
	MccJ25	*Escherichia coli*	a loop structure	Rat heart mitochondria	Inserted in the mitochondrial membrane and changed membrane permeability	[32]
	PETK	Cell-permeable Bak BH3 peptide	α-helices; a loop structure	Human T acute lymphoblastic leukemia CCRF-CEM cells	Combined with Bak and promoted MOMP	[33]
Inner mitochondrial membrane (IMM)	Cecropin A–melittin Hybrid Peptides	Artificial peptide	——	Rat liver mitochondria	MIMP was changed, and the IMM was destructed	[34]
	KLA	Artificial peptide	rich in Arg	Red blood cell (RBC) of Sprague Dawley rats	MIMP was changed	[35]
Mitochondrial permeability transition pore (MPTP)	BBJX	*Brevibacillus laterosporus* JX-5	——	Human U-937, mouse spleen cells	Opened the MPTP and stimulated the production of ROS	[14]
	BIRD-2	BH4 domain-targeting peptide	——	The SU-DHL-4 and KARPAS-422 DLBCL cell lines	Provoked mitochondrial Ca^2+^ overload followed by sustained MPTP opening	[36]
	BMAP-28	Bovine	α-helices	The U937 and K562 cell lines	Caused depolarization of the IMM and released Cyt *c*	[37]
	CGA-N12	Chromogranin-derived peptide	α-helices	*Candida tropicalis*	The accumulation of ROS in mitochondria causes MPTP to remain open	[38]
	Iturin	*Bacillus subtilis*	a loop structure	Myelogenous leukemia cells K562	Caused ROS burst and upregulated expression of Cyt *c*, Bax, and Bad, together with downregulated expression of Bcl-2	[13]
	Mt6-21DLeu	*Musca domestica* antifungal peptide MAF-1A derivative	α-helices	*Candida albicans*	Excess ROS was produced, followed by opening of the MPTP	[39]
	Peptide B11	*Litopenaeus vannamei* hemocyanin-derived peptide	α-helices	Human cervical cancer cells (HeLa), human hepatocellular carcinoma (HepG2)	Mitochondrial membrane potential was lost	[40]
	Scolopendin	*Scolopendra subspinipes mutilans*	α-helices	*Candida albicans*	Ca^2+^ homeostasis, membrane potential, and Cyt *c* levels were disrupted in mitochondria	[41]
	Scyreprocin	Mud crab *Scylla paramamosain*	rich in Lys	Human non-small-cell lung cancer NCI-H460 cells	Induced the generation of ROS and led to Ca^2+^ release	[42]
	Surfactin	*Bacillus subtilis*	a loop structure	Human OSCC cell linesSCC4/SCC25	Induced mitochondrial depolarization and mitochondrial-derived ROS production, Cyt *c* release	[12]
	TP3	*Oreochromis niloticus*	α-helices	OS MG63 cells	Excess ROS was produced, followed by sustained MPTP opening	[43]
	TP4	*Oreochromis niloticus*	α-helices	MCF-7 and A549 cell lines	Colocalized with ANT2 and regulated MPTP	[44]

MOMP: mitochondrial outer membrane permeability; MIMP: mitochondrial inner membrane permeability; Cyt *c*: cytochrome *c*; ROS: reactive oxygen species.

## Data Availability

Not applicable.

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
