# Peer review of "Antimicrobial Peptides Mediate Apoptosis by Changing Mitochondrial Membrane Permeability"

_ijms, 2022, doi:10.3390/ijms232112732_

Round 1
Reviewer 1 Report
There seems to be way too many small/grammatical issues within the text in order for the reader to be able to understand the presented ideas and concepts (see the minor points below).
1) Please change "(Y.Han): (optional; include country code; if there are multiple corresponding authors, add author initials)" to "(Y.Han)" (line 7).
2) Please replace "usually cause" with "cause" (line 11).
3) Please change "antimicrobial peptide-mediated" to "AMP-mediated" (line 12).
4) The permeability of "the mitochondrial outer membrane, mitochondrial inner membrane, and mitochondrial permeability transition pore" mentioned in "This paper reviews the molecular mechanisms of antimicrobial peptide-mediated apoptosis by changing the permeability of the mitochondrial membrane from three pathways: the mitochondrial outer membrane, mitochondrial inner membrane and mitochondrial permeability transition pore" seem to be referring to identical processes of MOMP, MIMP, and MPT mentioned in the following sentence "The roles of AMPs are also discussed in mitochondrial outer membrane permeability (MOMP), mitochondrial inner membrane permeability (MIMP), mitochondrial permeability transition (MPT) and apoptosis" (line 14). Please remove this redundancy.
5) Please replace "antimicrobial peptides" with "AMPs" (lines 17, 18, 30, 32, 40, 41, 55, 68, 72, 132, 135, 139, 171, 281, 284, 285).
6) Please change "Apoptosis" to "apoptosis" (line 21).
7) Please replace "usually" with "typically" (line 31).
8) Please replace "cells, but" with "cells but" (line 33).
9) Please change "the" to "an" (line 35).
10) Please replace "antimicrobial peptide" with "AMP" (lines 35, 70).
11) Please change "clinic" to "clinical" (line 36).
12) "Therefore, according to the mechanism of antimicrobial peptides in the treatment of tumors and infectious diseases, elucidating the way antimicrobial peptides exert their therapeutic effects is so that amp is screened more efficiently to develop therapeutic agents" (line 40) does not seem to make sense for the following reasons:
a) It is not clear why "tumors and infectious diseases" and not other pathophysiological conditions are mentioned as the target of AMP therapy?
b) It is not clearly why is "amp screened more efficiently to develop therapeutic agents"?
c) It is also not exactly clear what does "amp is screened" refer to?
d) "Therefore, according to the mechanism of antimicrobial peptides in the treatment of tumors and infectious diseases" and "elucidating the way antimicrobial peptides exert their therapeutic effects is so that amp is screened more efficiently to develop therapeutic agents" do not seem to logically fit each other's context.
13) Please replace "amp" with "AMP" (line 42).
14) Please format "Table 1" using the same font size consistent with the rest of the text (line 46).
15) "important role" appears twice in "Mitochondria, is a place for energy production in cells, playing an important role in regulating cell growth and death [11]. The integrity of the mitochondrial membrane plays an important role in mitochondrial apoptosis and even cell apoptosis, changes in mitochondrial membrane permeability can directly activate the mitochondrial apoptosis pathway" (line 46). Please fix.
16) Please change "Mitochondria, is a place for" to "Mitochondria is a site of" (line 46).
17) Please replace "descriptive of phenomena, such" with "descriptions of phenomena such" (line 51).
18) Please change "Only a" to "Only" (line 53).
19) Please replace "AMP" with something like "AMP therapy" (line 64).
20) Please change "discuss" to "discusses" (line 65).
21) "and provide support for the multitarget mechanism of antimicrobial peptides" does not seem to fit the preceding text in "The effect of amp during apoptosis would be analyzed from the perspective of MOMP, MIMP and MPT, and provide support for the multitarget mechanism of antimicrobial peptides" (line 66).
22) Please replace "amp during apoptosis would" with "AMPs during apoptosis will" (line 66).
23) It is not clear what the authors mean by "development" and "application" in "Some structural features of AMPs targeting the mitochondrial membrane are also discussed, which is aimed to elucidate the therapeutic mechanism and development as well as application of antimicrobial peptide" (line 68)?
24) Please change "peptide" to "peptides" (line 70).
25) Please define abbreviation for "Cytc" (LL-37), "RBC" (KLA), "cyt c" (BMAP-28), "MPTP" (CGA-N12) in Table 1.
26) Please merge abbreviation for "Cytc" and "cyt c" into only one instance in Table 1.
27) It is not clear what the authors mean by "The expression of Bax and Bcl-2 were regulated" in "The expression of Bax and Bcl-2 were regulated and changed the permeability of mitochondrial membrane" in Table 1 (17BIPHE2)?
28) It is also not clear what the authors mean by "ROS are gathered" in "ROS are gathered and followed by opening of the MPTP" in Table 1 (CGA-N12)?
29) From "It is able to insert in the mitochondria membrane, changes membrane permeability" is not clear which mitochondrial membrane are the authors referring to in Table 1 (MccJ25)?
30) "and followed by" might not be grammatically correct in Table 1 (CGA-N12, BIRD-2, TP3, Mt6-21DLeu). Please revise.
31) Please replace "Combined with Bak, and promoted" with "Combined with Bak and promoted" (PETK), "Released Cytc, and promoted" with "Released Cytc and promoted" (LL-37, LTX-315), "mitochondrial outer membrane permeability" with "MOMP" (PETK, LL-37, LTX-315), "Mitochondrial inner membrane permeability" with "MIMP" (KLA, Cecropin-A–Melittin Hybrid Peptides), "Cecropin-A–Melittin" with "Cecropin A–Melittin" (Cecropin-A–Melittin Hybrid Peptides), "Human cervical" with "Human cervical cancer cells" (Peptide B11), "It causes" with "Caused" (BMAP-28), "calcium ion" with "Ca2+" (Scolopendin), "mitochondria," with "mitochondria" (Scolopendin), "It is" with "Is" (MccJ25), "It colocalizes with ANT2,and" with "Colocalized with ANT2," (TP4), "Cell" with "cell", "rScyreprocin" with "Scyreprocin" (rScyreprocin), "It induced" with "Induced" (rScyreprocin, surfactin), "surfactin" with "Surfactin", "iturin" with "Iturin", "It causing" with "Caused" (iturin), "It opens the MPTP and stimulating" with "Opened the MPTP and stimulated" in Table 1.
32) Please format "2+" in "Ca2+" using superscript in Table 1 (BIRD-2, rScyreprocin).
33) Please change "Cytochrome c and Apoptosis Inducing Factor" to "cytochrome c and apoptosis inducing factor" (line 75).
34) It is not clear what the authors mean by "mitochondrial membrane gap" in "MOMP refers to the process by which proteins (Cytochrome c and Apoptosis Inducing Factor) existing in the mitochondrial membrane gap enter the cytoplasm through the outer membrane, activate the apoptotic pathway or trigger inflammatory reactions, causing mitochondria and cell apoptosis" (line 75)?
35) From "MOMP refers to the process by which proteins (Cytochrome c and Apoptosis Inducing Factor) existing in the mitochondrial membrane gap enter the cytoplasm through the outer membrane, activate the apoptotic pathway or trigger inflammatory reactions, causing mitochondria and cell apoptosis" (line 75) is not clear to which mitochondrial membrane are the authors referring to and what is the difference between "mitochondria and cell apoptosis"?
36) Please replace "outer membrane" with "mitochondrial outer membrane" (lines 77, 95, 184).
37) Please change "mitochondria" to "mitochondrial" (line 78).
38) Please replace "Mitochondrial" with "mitochondrial" (line 79).
39) Please change "Caspase3" to "caspase-3" (lines 86, 133).
40) "These artificially modified antimicrobial peptides take the outer membrane of mitochondria as the specific target which are good candidate drugs for the treatment of cancer and infectious diseases" (line 86) does not seem to be grammatically correct with respect to "take the outer membrane of mitochondria as the specific target". Please rephrase.
41) "which are good candidate drugs for the treatment of cancer and infectious diseases" does not seem to fit "These artificially modified antimicrobial peptides take the outer membrane of mitochondria as the specific target which are good candidate drugs for the treatment of cancer and infectious diseases" (line 86).
42) It is not clear what the authors mean by "mitochondrial membrane gap" in "LL-37 can accumulate in the mitochondria of human osteogenic MG63 cells and promote the release of apoptosis inducing factor (AIF) and cytochrome c (Cyt c) from the mitochondrial membrane gap, while cytochrome c oxidase IV (COX IV) located on the inner membrane of mitochondria was not detected" (line 88)?
43) Please replace "apoptosis inducing factor (AIF) and cytochrome c (Cyt c)" with "cytochrome c (Cyt c) and apoptosis inducing factor (AIF)" (line 90).
44) Please replace "COX IV" with "COXIV" (line 91).
45) Please change "on" to "in" (line 91).
46) Please replace "related" with "attributed" (line 99).
47) "It is possible that during the process of MOMP, mitochondria initiate MPTP and MIMP, and then destroy the structure of mitochondria together" (line 100) is not semantically correct as it is hard to imagine that "mitochondria" can "destroy the structure of mitochondria". Please revise.
48) Please change "and HN interacts" to "interact" (line 103).
49) Please replace "that,in the process of" with "that during" (line 105).
50) Please change "matrix" to "mitochondrial matrix" (line 106).
51) Please replace "Caspase" with "caspase" (line 108).
52) Please change "in the process of" to "during" (line 109).
53) Please replace "inflammation(Error! Reference source not found.)" with "inflammation" (line 110).
54) Please format "2+" in "Ca2+" using superscript in Figure 1.
55) Please change "mitochondria outer membrane" to "mitochondrial outer membrane", "Caspase3" to "caspase-3", "Procaspase9" to "procaspase-9", "Apoptotic" to "apoptosis", "Apoptotic bodies" to "apoptotic bodies", "Apaf1" to "Apaf-1", "imflammation" to "inflammation" in Figure 1.
56) Please copy and paste the "Apaf1" caption and icon next to the arrow leading from "Cyt c" from the lower to the upper section of Figure 1.
57) Please relabel "(1) MOMP", "(3) MPT", "(2) MIMP" to "(1) MOMP", "(2) MPT", "(3) MIMP" in Figure 1 and the respective legend.
58) Please replace "Cytochrome C" with "Cytochrome c" (lines 113).
59) Please change "SMAC" to "Smac" (lines 114, 117, 123).
60) Please replace "Cytochrome" with "cytochrome" (lines 115).
61) Please change "Apaf1 and Procaspase9" to "Apaf-1 and procaspase-9" (line 115).
62) Please replace "caspase3" with "caspase-3" (line 116).
63) Please change "Caspase 3" to "Caspase-3" (line 116).
64) Please replace "caspase" with "caspases" (line 118).
65) Please change "CGAS-STING" to "cGAS-STING" (line 118).
66) Please replace "reactive oxygen species" with "ROS" (line 120).
67) Please format "2+" in "Ca2+" using superscript (line 120).
68) Please replace "In the process of permeability transition" with "During MPT" (line 122).
69) Please change "cyt C" to "Cyt C" (line 123).
70) Please replace "are also released, which also induce apoptosis in a caspase3-dependent" with "are released, inducing apoptosis in a caspase-3-dependent" (line 123).
71) It is not clear what the authors mean by "the above evidence" in "From the above evidence, AMPAs play an apoptotic role by inducing the initiation of MOMP." (line 125)?
72) Please change "AMPAs" to "AMPs" (line 125).
73) "One way is to directly participate in MOMP to promote cell apoptosis based on the specific structure of apoptotic proteins regulating MOMP, which is commonly seen in the derived peptides of the pro-apoptotic protein family" (line 126) is puzzling because of the following reasons:
a) It is not exactly clear what the authors refer to as "One way".
b) "is to directly participate in MOMP" lacks a concrete noun. What entity directly participates in MOMP?
c) The role how "specific structure of apoptotic proteins" regulates MOMP is elusive at this topic has not been specifically discussed in the text.
d) "which is commonly seen in the derived peptides of the pro-apoptotic protein family" does not seem to fit the sentence.
e) It is not clear what the authors mean by "derived peptides of the pro-apoptotic protein family"?
Please fix this sentence.
74) Similarly, "The other way is to regulate the expression of apoptotic proteins that can initiate or prevent MOMP to indirectly regulate MOMP and induce subsequent apoptosis" (line 128) is puzzling for the following reasons:
a) It is not exactly clear what the authors refer to as "The other way".
b) "that can initiate or prevent MOMP to indirectly regulate MOMP" does not make sense since it expresses the same meaning twice (initiate or prevent MOMP = regulate MOMP).
c) It is hard to imagine that "apoptotic proteins that" "prevent MOMP" induce "subsequent apoptosis".
75) "If caspases are absent, they can also trigger inflammation with the release of pro-apoptotic factors" (line 130) does not seem to make sense as it is hard to imagine that caspases "trigger inflammation" when they are absent.
76) Please replace "If caspases are absent" with "In the absence of caspases" (line 130).
77) From "Therefore, there is no evidence that antimicrobial peptides promote signaling during apoptosis, although increased Caspase3 activation in the apoptotic pathway was detected" (line 131) is not explicitly clear whether "increased Caspase3 activation in the apoptotic pathway was detected" following AMP treatment?
78) It is not exactly clear what the authors mean by "this" and "apoptotic chain reaction" in "However, it is not clear whether this is caused by the apoptotic chain reaction caused by antimicrobial peptides or by the increased expression of their genes caused by antimicrobial peptides themselves" (line 133)?
79) "their genes caused by antimicrobial peptides themselves" does not seem to make sense in "However, it is not clear whether this is caused by the apoptotic chain reaction caused by antimicrobial peptides or by the increased expression of their genes caused by antimicrobial peptides themselves" (line 133) since "their" and "antimicrobial peptides" refer to the same entity (AMPs).
Author Response
Dear review 1:
We are grateful for your valuable suggestions and recognition. Thank you for spending a lot of valuable time patiently reviewing the manuscript. At the same time, we have received a lot of help from you. We have solved the problem you raised one by one, all our changes of manuscript are marked up using the “Track Changes” function of MS Word. May happiness and health be with you always. We have attached a detailed response to the document for your review.
Yours sincerely
Dr. Yuzhu Han

Reviewer 2 Report
In this review, " Antimicrobial peptides mediate apoptosis by changing mitochondrial membrane permeability”, Hongji WANG et al., systematically described the ways in which AMPs alter mitochondrial membrane permeability. The effect of the antimicrobial peptides during apoptosis has been analyzed from the point of view of mitochondrial outer membrane permeability (MOMP), mitochondrial inner membrane permeability (MIMP), and mitochondrial permeability transition (MPT), to support the multitarget mechanism of antimicrobial peptides.
The comments and suggestions for this manuscript are as follows-
1. Page 5 line 112; shows figure-1, and Page 9 line 288 also shows figure-1. The authors must correct it and the corresponding text/reference in the main manuscript, wherever required.
2. The details of figure(s) legends are required.
3. Page5 line 110, page 6 line 153, and page 7 line 222. “Error! Reference source not found.” What does it mean? The author must correct it and add proper references.
4. Page 6 lines 138-140, “we temporarily concluded that most antimicrobial peptides promote apoptosis by an indirect way by promoting MOMP”. What is the meaning of temporarily concluded? The Autor must replace sentences with proper scientific wording.
5. Page 8 lines 268-270. Recheck the sentence and formatting.
6. The introduction and the main body of the manuscript are typical textbook types. This is lacking intellectual input from authors. The author must provide a comprehensive introduction and conclusion with proper references.
7. All authors need to read and pay attention to review this manuscript before re-submission of the revised version.
Author Response
Dear review 2,
We appreciate your rigorous attitude, professional evaluation, and constructive suggestions. Thank you for spending a lot of valuable time patiently reviewing manuscripts. We have solved the problem you raised one by one, all our changes of manuscript are marked up using the “Track Changes” function of MS Word. All authors have read and reviewed the revised manuscript. May happiness and health be with you always.
With kindest regards,
Yours sincerely
In this review, " Antimicrobial peptides mediate apoptosis by changing mitochondrial membrane permeability”, Hongji WANG et al., systematically described the ways in which AMPs alter mitochondrial membrane permeability. The effect of the antimicrobial peptides during apoptosis has been analyzed from the point of view of mitochondrial outer membrane permeability (MOMP), mitochondrial inner membrane permeability (MIMP), and mitochondrial permeability transition (MPT), to support the multitarget mechanism of antimicrobial peptides.
The comments and suggestions for this manuscript are as follows-
1. Page 5 line 112; shows figure-1, and Page 9 line 288 also shows figure-1. The authors must correct it and the corresponding text/reference in the main manuscript, wherever required.
A: Dear reviewer, we want to show different emphases in Figures 1 and 2. In figure 1, while explaining the mechanism of action of antimicrobial peptides, we describe how the three pathways that alter mitochondrial membrane permeability are initiated and how they function in detail. On the basis of the mechanism of altered mitochondrial membrane permeability, figure 2 illustrates how AMPs are linked to the three pathways that alter mitochondrial membrane permeability.
In addition, figures 1 and 2 are illustrated on page 5, line 112, and page 9, line 288. We have made the appropriate modifications to page 9, line 288, based on the content of the figure.
Manuscript: “AMPs can induce MOMP and MPT. In the presence of caspase, the caspase cascade can be triggered, leading to apoptosis. In the absence of caspase, the CGAS-STING signaling pathway leads to inflammation and cell necrosis. Because MOMP, MIMP and MPT are closely related, MOMP can lead to MIMP, and MPT interacts with MOMP and MIMP. Therefore, AMPs can also affect one aspect and then indirectly affect the other two.”
Correction: “On the one hand, AMPs can induce MOMP and MPT directly. On the other hand, because MOMP, MIMP and MPT are closely related, AMPs can indirectly induce MIMP. In a word, AMPs can affect one aspect and then in-directly affect the other two, causing inflammation or apoptosis in cells (page 9 lines 298-301).”
2.The details of figure(s) legends are required.
A: The legend has been modified and added after the name of the figure 1(page 5 lines 129-138) and 2 (page 9 lines 298-301).
3.Page5 line 110, page 6 line 153, and page 7 line 222. “Error! Reference source not found.” What does it mean? The author must correct it and add proper references.
A: Dear reviewer, after carefully reading the manuscript, we found that there is no “Error! Reference source not found.” Lines 110 on page 5 (line 128) and 153 on page 6 (line 168) respectively cite the references [39] and [46]. In addition, line 22 on page 7 does not cite relevant literature.
References:
- Giampazolias, E.; Zunino, B.; Dhayade, S.; Bock, F.; Cloix, C.; Cao, K.; Roca, A.; Lopez, J.; Ichim, G.; Proïcs, E.; et al. Mitochondrial Permeabilization Engages NF-ΚB-Dependent Anti-Tumour Activity under Caspase Deficiency. Nat. Cell Biol. 2017, 19, 1116–1129, doi:10.1038/ncb3596.
- Riley, J.S.; Quarato, G.; Cloix, C.; Lopez, J.; O’Prey, J.; Pearson, M.; Chapman, J.; Sesaki, H.; Carlin, L.M.; Passos, J.F.; et al. Mitochondrial Inner Membrane Permeabilisation Enables MtDNA Release during Apoptosis. EMBO J. 2018, 37, e99238, doi:10.15252/embj.201899238.
4.Page 6 lines 138-140, “we temporarily concluded that most antimicrobial peptides promote apoptosis by an indirect way by promoting MOMP”. What is the meaning of temporarily concluded? The Autor must replace sentences with proper scientific wording
A: We have corrected the error here, and the revised sentence is on page 6, line 150.
Correction: “for now, we can draw a conclusion that most antimicrobial peptides promote apoptosis by an indirect way by promoting MOMP.”
5.Page 8 lines 268-270. Recheck the sentence and formatting.
A: We have checked and modified the sentence formatting. The revised sentence is on page 8, lines 279-283.
Correction: “The mechanism of apoptosis caused by the three pathways has been confirmed by many experiments: one is apoptosis mediated by a caspase protein cascade response triggered by the release of Cytc [74]; the other is the caspase-independent pathway, mediated by apoptosis-inducing factor (AIF) or inflammatory signals such as mtDNA [75, 76].”
References
- Oberst, A.; Bender, C.; Green, D.R. Living with Death: The Evolution of the Mitochondrial Pathway of Apoptosis in Animals. Cell Death Differ. 2008, 15, 1139–1146, doi:10.1038/cdd.2008.65.
- Scovassi, A.I.; Soldani, C.; Veneroni, P.; Bottone, M.G.; Pellicciari, C. Changes of Mitochondria and Relocation of the Apoptosis-Inducing Factor during Apoptosis. Ann. N. Y. Acad. Sci. 2009, 1171, 12–17, doi:10.1111/j.1749-6632.2009.04707.x.
- Zhou, L.; Zhang, Y.-F.; Yang, F.-H.; Mao, H.-Q.; Chen, Z.; Zhang, L. Mitochondrial DNA Leakage Induces Odontoblast Inflammation via the CGAS-STING Pathway. Cell Commun. Signal. 2021, 19, 58, doi:10.1186/s12964-021-00738-7.
6.The introduction and the main body of the manuscript are typical textbook types. This is lacking intellectual input from authors. The author must provide a comprehensive introduction and conclusion with proper references.
A: We appreciate your constructive suggestions. We reorganized the writing ideas and carefully checked the full text. We have earnest revised the introduction and conclusion, corrected the improper points in the main body and added legends for Figures 1 and 2.
7.All authors have read and reviewed the revised manuscript.
Reviewer 3 Report
The review on the topic "Antimicrobial peptides mediate apoptosis by changing mitochondrial membrane permeability" is devoted to the description of the few examples of the action of AMPs of various origins that penetrate the target cell and have a targeted effect on this cell organelle. The uniqueness of this work lies in the fact that it summarizes the available information on the biological activity (antitumor and antifungal) of a number of known natural and synthetic antimicrobial peptides, many of which, at sub-inhibitory concentrations, realize the cell penetration effect. Moreover, further understanding of exactly how the suppressive effect of the peptide is realized at the molecular level inside the cell is of the highest interest. In this regard, we can conclude that this mini-review will be extremely useful for scientists working in many areas associated with the study of antibiotics. There are no questions or comments on the content, except that it would be advisable to present Table 1 not in the "Introduction" section, but in a separate block devoted to describing the variety of forms of peptides with antifungal and antitumor properties.
Author Response
Dear reviewer 3
Thank you for your recognition of our work, kind comments and helpful suggestions. We have carefully revised our disease note according to the comments and suggestions, all our changes of manuscipt are marked up using the “Track Changes” function of MS Word. With best wishes for happiness in your life and work.
With kindest regards,
Yours Sincerely
Dr. Yuzhu Han
Q: There are no questions or comments on the content, except that it would be advisable to present Table 1 not in the "Introduction" section, but in a separate block devoted to describing the variety of forms of peptides with antifungal and antitumor properties.
A: Thanks for your unique insight. To enrich the review, we have added on page 2 about antimicrobial peptides and apoptosis. Table 1 contains all the antitumor and antifungal peptides mentioned in this review. Therefore, the relationship between each AMP and apoptosis was not detailed here. Instead, we want to highlight that AMPs are closely related to apoptosis, in which the signal transduction of the mitochondrial pathway is the main pathway.
In addition, to make this paper more perfect, we fixed the introduction, the wrong language in the manuscript, and added legends for Figures 1 and 2.
Round 2
Reviewer 1 Report
Major points:
1) The resolution of Figures 1 and 2 is poor and many of the detailed features are now pixelated. Please increase its size and/or pixel density to achieve the level of sharpness for this figure to a similar standard as seen in the last round of revision.
2) Could the authors please design a new figure illustrating the molecular composition of MPTP? This would help the readers to better understand the developed concept of permeability transition.
Minor points:
1) Please change "were" to "are" (line 15).
2) Please replace "AMPs" with "AMP" (lines 31, 35, 119).
3) Please change "treatment, such" to "treatment such" (line 32).
4) Please replace "understood[10, 11]" with "understood [10, 11]" (line 38).
5) Please change "However" to something like "Nevertheless" (line 39).
6) It is not exactly clear what the auhtors mean by "efficiency" in "Therefore, sorting out the mechanism of AMPs that treat tumors and infectious diseases by the intracellular pathway is convenient to screen AMPs with better efficiency and develop them as therapeutic drugs" (line 40)? Efficiency in terms of what?
7) Please replace "that treat" with "for the treatment of" (line 40).
8) Please change "with" to "for" (line 42).
9) Please replace "AMPs could" with "AMPs" (line 44).
10) Please change "functions[12-14]" to "functions [12-14]" (line 45).
11) Please replace "descriptive" with "descriptions" (line 107).
12) Please change "[19], and" to "[19] and" (line 110).
13) "and provide support for the multitarget mechanism of AMPs" does not seem to fit "The effect of AMPs during apoptosis will be analyzed from the perspective of MOMP, MIMP and MPT, and provide support for the multitarget mechanism of AMPs" (line 121). Please revise.
14) "Some structural features of AMPs targeting the mitochondrial membrane are also discussed, which is aimed to analyze the therapeutic mechanism of AMPs and to modify AMPs using their characteristic structures" (line 123) does not seem to make sense for the following reasons:
a) "which is aimed to analyze the therapeutic mechanism of AMPs and to modify AMPs using their characteristic structures" does not seem to perfectly fit the sentence.
b) "modify AMPs using their characteristic structures" might not be semantically correct as it is not clear how "characteristic structures" could be used to "modify AMPs".
c) It is not clear whether the authors are referring to outer or inner mitochondrial membrane?
15) Please replace "regulated, active and programmed death process" with "regulated and active programmed death process" (line 127).
16) Please replace "homeostasis[22]" with "homeostasis [22]" (line 128).
17) Please change "mitochondrial" to "the mitochondrial" (line 129).
18) Please replace "pathway[14, 23]" with "pathway [14, 23]" (line 130).
19) Please change "membranes[25]" to "membranes [25]" (line 131).
20) Please replace "it" with "they" (lines 132, 134).
21) Please change "bacteria[26]" to "bacteria [26]" (line 134).
22) Please replace "metabolism[7]" with "metabolism [7]" (line 135).
23) It is not clear whether the authors are referring to outer or inner mitochondrial membrane in "In fact, we observed that AMPs that enter the cell and exert their effects are often able to achieve programmed cell death by mitochondrial swelling, mitochondrial membrane rupture, and stimulating the activation of apoptotic markers" (line 135)?
24) Please change "rupture, and" to "rupture and" (line 137).
25) Please change "in the process of mitochondrial pathway" to "during mitochondrial apoptosis" (line 139).
26) Please replace "explain" with "describe" (line 140).
27) Please replace "AMPs induced" with "AMP-induced" (line 140).
28) Please change "Table 1" to "Table 1." (line 163).
29) It is not clear whether the authors are referring to outer or inner mitochondrial membrane in "Effect of some AMPs on the mitochondrial membrane" (line 163)?
30) Please replace "red" with "Red" (KLA), "changed, and" with "changed and" (Cecropin A–Melittin Hybrid Peptides), "burst, and indicated by the" with "burst and" (Iturin), "Bad, together" with "Bad together" (Iturin), "Bcl-2." with "Bcl-2" (Iturin), "produced and" with "produced," (TP3), "opening." with "opening" (TP3), "Excess ROS is produced" with "Excess ROS was produced" (Mt6-21DLeu), "membrane, releases Cyt c" with "membrane and released Cyt c" (BMAP-28), "ANT2, and" with "ANT2 and" (TP4), "in the mitochondria" with "in the mitochondrial" (MccJ25), "Induced the generation" with "Induced the generation" (Scyreprocin), "ROS, and led to" with "ROS and led to" in Table 1.
31) Please format "Brevibacillus laterosporus" (BBJX), "Oreochromis niloticus" (TP3, TP4), "Musca domestica" (Mt6-21DLeu), "Litopenaeus vannamei" (Peptide B11), "Scolopendra subspinipes mutilans" (Scolopendin), "Scylla paramamosain" (Scyreprocin) using italics in Table 1.
32) Please sort all AMPs presented in Table 1 in alphabetical order.
33) Please format "MOMP: mitochondrial outer membrane permeability; MIMP: mitochondrial inner membrane permeability; Cyt c: cytochrome c; ROS: reactive oxygen species" (line 192) using consistent font type and size.
34) Please change "MOMPs" to "MOMP" (lines 194, 673).
35) Please replace "apoptosis inducing" with "apoptosis-inducing" (lines 195, 209).
36) Please change "causing" to "thereby causing" (line 198).
37) Please replace "PETK" with "PETK, which" (line 203).
38) It is not clear what the authors mean by "artificially modified AMPs" in "These artificially modified AMPs targeting the mitochondrial outer membrane are good candidate drugs for the treatment of tumors and infectious diseases" (line 206)?
39) Although there seem to be no prior mention of "artificially modified AMPs" preceding to "These artificially modified AMPs targeting the mitochondrial outer membrane are good candidate drugs for the treatment of tumors and infectious diseases" (line 206), these are being referred to as "These". Please fix.
40) "while Cyt c oxidase IV (COXIV) located in the inner membrane of mitochondria was not detected" does not seem to logically fit in "LL-37 can accumulate in the mitochondria of human osteogenic MG63 cells and promote the release of cytochrome c (Cyt c) and apoptosis inducing factor (AIF) between the outer and inner mitochondrial membranes, while Cyt c oxidase IV (COXIV) located in the inner membrane of mitochondria was not detected" (line 208). What is the reason for stressing out that COXIV could not be detected in the context of Cyt c and AIF released between the outer and inner mitochondrial membranes? If indeed authors mean to say that COXIV degradation leads to Cyt c and AIF release, this has to be explicitly stated.
41) Please provide reference for ""LL-37 can accumulate in the mitochondria of human osteogenic MG63 cells and promote the release of cytochrome c (Cyt c) and apoptosis inducing factor (AIF) between the outer and inner mitochondrial membranes, while Cyt c oxidase IV (COXIV) located in the inner membrane of mitochondria was not detected" (line 208)".
42) Please change "its" to "the" (line 214).
43) It is not clear what the authors mean by "the mitochondrial crest structure" in "Notably, the mitochondrial crest structure was damaged during this process, which may be attributed to the increase in intracellular reactive oxygen species (ROS) and Ca2+ concentrations caused by 17BIPHE2" (line 218)?
44) Please replace "concentrations" with "concentration" (line 220).
45) Please change "during the process of" to "during" (line 220).
46) Please replace "occur in mitochondria at the same time, and the three parties" with "simultaneously occur in mitochondria and the three processes" (line 221).
47) Please change "remove" to "prevent" (line 265).
48) Please replace "some" with "several" (line 266).
49) Please replace "inflammation(Figure 1)" with "inflammation (Figure 1)" (line 266).
50) Please center "ROS", "Ca2+", and "others" properly within the blue circles in Figure 1.
51) Please enlarge the blue ellipse to fully cover the "membrane potential" caption in Figure 1.
52) Please enlarge the dark orange ellipse to fully cover the "inflammation" caption in Figure 1.
53) Please change "apoptotic" on the orange background to "apoptosis" in Figure 1.
54) Please replace "Procaspase-9" with "procaspase-9" (line 270).
55) Please change "mtDNA" to "mitochondrial DNA (mtDNA)" (line 274).
56) Please replace "inner and outer" with "outer and inner" (line 276).
57) It is not clear what the authors mean by "the above research" in "According to the above research, we can draw a conclusion" (line 279)? A better way of referencing to previous research would be by using a citation.
58) "According to the above research, we can draw a conclusion" (line 279) sounds rather vague.
59) Please change "structure" to "framework" (line 281).
60) Please replace "they" with "AMPs" (line 284).
61) "This section will be covered in detail in MIMP" (line 285) is also a vague statement.
62) Please change "the process from the" to "the" (line 288).
63) Please replace "to" with "to trigger" (line 288).
64) Please change "caused" to "caused directly" (line 288).
65) Please replace "expression of caspase genes caused by AMPs" with "AMP-induced expression of caspase genes" (line 420).
66) Please change "sufficient" to "thorough" (line 423).
67) Please replace "ndireclyt" with "indirectly" (line 424).
68) It is not clear what the authors mean by "substances" in "MIMP is considered to be a process that occurs after MOMP, and substances in the mitochondrial matrix are released into the cytoplasm across the mitochondrial membrane" (line 426)?
69) It is also not clear whether the authors are referring to outer or inner mitochondrial membrane in "MIMP is considered to be a process that occurs after MOMP, and substances in the mitochondrial matrix are released into the cytoplasm across the mitochondrial membrane" (line 426)?
70) Please change "continues" to "continue" (line 428).
71) Please replace "pores formed in the outer membrane will also" with "formed pores will" (line 429).
72) It is not clear what the authors mean by "ions blocked by the inner membrane of the mitochondria" in "Of course, if the degree of MIMP is mild, some of the ions blocked by the inner membrane of the mitochondria are allowed to reach the outside, causing changes in membrane potential" (line 432)? Are they referring to calcium ions? How can ions be blocked by the inner membrane? Do the authors mean to say "the passage of ions"?
73) It is not exactly clear what the authors mean by "outside" in "Of course, if the degree of MIMP is mild, some of the ions blocked by the inner membrane of the mitochondria are allowed to reach the outside, causing changes in membrane potential" (line 432)? Please refer to mitochondrial compartments as the mitochondrial matrix, mitochondrial inner membrane, intermembrane space, or mitochondrial outer membrane.
74) Please change "determine that" to "determine" (line 439).
75) Please replace "in the process of" with "during" (line 440).
76) "In addition to the classical pathway from MOMP to MIMP" does not seem to make sense (line 443).
77) Please change "peptides, such as KLAKLAK2 (KLA), can" to "peptides such as KLAKLAK2 (KLA) can" (line 444).
78) Please replace "mitochondrial inner" with "mitochondrial" (lines 444, 446).
79) Please change "potential, but do not initiate" to "potential but without initiating" (line 445).
80) It is not clear whether the authors are referring to outer or inner mitochondrial membrane in "Moreover, the ability of KLA to penetrate the mitochondrial membrane is dependent on the mitochondrial inner membrane potential" (line 445)?
81) Please replace "whatever" with "any" (line 447).
82) Please change "will almost" to "will" (line 447).
83) Please replace "precursor" to "preceding event" or "preceding step" (line 449).
84) It is not clear which AMP are the authors referring to as "cecrosine-melittin" in "Early research on cecrosine-melittin hybrid short peptide found that it could pass through the mitochondrial inner membrane and allow the entry and exit of some substances, but it would destroy the structure of the mitochondrial inner membrane [51]" (line 449)? Moreover, "cecrosine-melittin" is nowhere to be found in reference [51].
85) It is also not clear what the authors mean by "substances" in "Early research on cecrosine-melittin hybrid short peptide found that it could pass through the mitochondrial inner membrane and allow the entry and exit of some substances, but it would destroy the structure of the mitochondrial inner membrane" (line 449)?
86) It is not exactly clear what the authors mean by "on the basis of MOMP" in "In a study of mitochondria-mediated apoptosis, mtDNA leakage occurred on the basis of MOMP after the treatment of cells with the BH3 mimic inhibitor ABT-737" (line 452)?
87) Please remove bold formatting from "emb" in "membrane" (line 455).
88) Please change "mitochondrial DNA" to "mtDNA" (line 459).
89) From "In either case, we need to continue to explore" (line 462) is not explicitly clear what the authors wish to "continue to explore"?
90) Please replace "[52- 54]" with "[52-54]." (line 469).
91) Please change "the MPTP" to "MPTP" (line 659).
92) It is not clear what the authors mean by "expansion of mitochondrial matrix solute osmotic pressure" in "If it is in a continuous open state, mitochondrial permeability transition (MPT) occurs, which can lead to rapid expansion of mitochondrial matrix solute osmotic pressure, rupture of the mitochondrial outer membrane, collapse of mitochondrial membrane potential, depletion of cell ATP, and ultimately cell necrosis or apoptosis" (line 659)? It is namely not clear how can osmotic pressure be expanded?
93) Please provide reference for "Dithiothreitol (DTT) can inhibit sulfhydryl oxidation on MPTP, and cyclosporin A (CsA) can interact with Cyp-D, both of which are typical MPTP opening inhibitors" (line 663).
94) The statement that DTT is an "MPTP opening inhibitor" is rather harsh in "Dithiothreitol (DTT) can inhibit sulfhydryl oxidation on MPTP, and cyclosporin A (CsA) can interact with Cyp-D, both of which are typical MPTP opening inhibitors" (line 663) since it is hard to imagine that DTT can inhibit MPTP opening specifically.
95) Please replace "and can" with "that" (line 670).
96) Please change "caused" to "causes" (line 674).
97) Please replace "and" with "and other" (line 674).
98) "and the changes in their impairment are mainly the result of a combination of factors" does not seem to fit "In addition, there are many factors affecting the opening of the MPTP, and the changes in their impairment are mainly the result of a combination of factors" (line 675) very well.
99) It is not clear what the authors mean by "their" in "In addition, there are many factors affecting the opening of the MPTP, and the changes in their impairment are mainly the result of a combination of factors" (line 675)?
100) Please either give example of some of the factors mentioned in "combination of factors" as part of "In addition, there are many factors affecting the opening of the MPTP, and the changes in their impairment are mainly the result of a combination of factors" (line 675).
101) Please change "ROS in mitochondrial and intracellular" to "mitochondrial and intracellular ROS" (line 677).
102) It is not clear what the authors mean by "it" in "On the other hand, it affects mitochondrial fission, leads to mitochondrial dysfunction and increases the probability of MPTP opening" (line 679)?
103) Analogously, it is not clear what the authors mean by "It" in "Finally, It leads to enhanced MPT in mitochondria, prompting activation of caspase3/9" (line 681)?
104) Please replace "It" with "it" (line 681).
105) Please change "MPT in mitochondria, prompting" to "MPT thereby prompting" (line 681).
106) Please replace "promotes the process of" with "facilitates" (line 682).
107) Please format "Litopenaeus vannamei" using italics (line 682).
108) Please replace "MPTP continuous" with "continuous MPTP" (line 686).
109) Please change "cells[63]" to "cells [63]" (line 689).
110) Please either remove "in the cells" from "After treatment with AMPs, Cyt c is released, ROS is increased, and the mitochondrial membrane potential is disrupted in the cells" (line 688) or specify the type of cells.
111) Please replace "All these results confirmed" with "Altogether, these results confirm" (line 689).
112) Please change "way" to "manner" (line 691).
113) Please replace "And" with "In addition," (line 691).
114) It is not clear what the authors mean by "And AMPs strengthened the necessary link between MPTP and MOMP" (line 691)? How treatment with AMPs can strengthened the link between MPTP and MOMP? Why is this link necessary?
115) It is not clear what the authors refer to as "potential changes" in "Although it can be inferred from potential changes that MIMP exists in the process of MPT, unfortunately, it is not clear whether MIMP is caused by the continuous opening of MPTP or whether MOMP also plays a role" (line 692)?
116) Please change "exists in the process of" to "occurs during" (line 693).
117) Please replace "unfortunately, it" with "it" (line 693).
118) "it is not clear whether MIMP is caused by the continuous opening of MPTP or whether MOMP also plays a role" does not make sense in "Although it can be inferred from potential changes that MIMP exists in the process of MPT, unfortunately, it is not clear whether MIMP is caused by the continuous opening of MPTP or whether MOMP also plays a role" (line 692) as "MIMP is caused by the continuous opening of MPTP" and "MOMP also plays a role" are not mutually exclusive. Please revise.
119) Please change "role" to "role in this process" (line 695).
120) Please provide references for "Tumor cells are not sensitive to their own apoptosis signals, and infectious diseases are faced with the problem of antibiotic abuse leading to drug resistance of pathogenic bacteria" (line 697).
121) "Tumor cells are not sensitive to their own apoptosis signals, and infectious diseases are faced with the problem of antibiotic abuse leading to drug resistance of pathogenic bacteria" (line 697) is rather puzzling as it is not clear why "Tumor cells are not sensitive to their own apoptosis signals" is mentioned in connection with "infectious diseases are faced with the problem of antibiotic abuse leading to drug resistance of pathogenic bacteria"?
122) "infectious diseases are faced with the problem of antibiotic abuse" does not seem to be semantically correct in "Tumor cells are not sensitive to their own apoptosis signals, and infectious diseases are faced with the problem of antibiotic abuse leading to drug resistance of pathogenic bacteria" (line 697)" as it is hard to imagine that "infectious diseases" can be faced with a problem.
123) Please replace "are also" with "are" (line 701).
124) Please change "modifications, such" to "modifications such" (line 704).
125) Please replace "protein fusion, glycosylation" with "glycosylation, protein fusion" (line 704).
126) Please change "treatment" to "therapeutic" (line 706).
127) "To enhance the utilization of AMPs, those AMPs that cannot enter cells to play a role can be modified to make them penetrate the cell membrane" (line 706) is puzzling for the following reasons:
a) It is not clear what role AMPs are supposed to play?
b) The context of why the authors are referring to "AMPs that cannot enter cells" is missing.
128) Please replace "make them penetrate the cell membrane" with something like "render them membrane permeant" (line 707).
129) From "For example, a series of mitochondrial penetrating peptides are prepared by modification with arginine for the delivery of AMPs" (line 708) is not explicitly clear whether AMPs are delivered into mitochondria using arginine-modified mitochondrial penetrating peptides or whether the arginine is modified on AMPs themselves?
130) Please change "design known" to "design" (line 746).
131) Please replace "or" with "or to" (line 746).
132) "AMPs widely exist in various organisms, and there are many kinds, so it is difficult to use them effectively" (line 751) is puzzling for the following reasons:
a) It is not explicitly clear whether the authors mean to express that there are "many kinds" of organisms or AMPs?
b) The reason why "it is difficult to use" AMPs "effectively" in the context of the sentence is not clear. Please explain in the text.
133) "Linking the specific mechanism and structures of AMPs can not only effectively study the mechanism of action of known AMPs on mitochondrial membranes but also clarify the mechanism of promoting apoptosis" (line 753) is not semantically correct as it is hard to imagine that linking "the specific mechanism and structures of AMPs" can study.
134) Please change "and structures" to "with structure" (line 754).
135) Please replace "also" with "also can help to" (line 755).
136) "While choosing safer and more selective AMPs, it also weakens the problem of AMPs in clinical application to a certain extent" (line 756) does not seem to be grammatically correct with respect to "While choosing safer and more selective AMPs, it also weakens". Please rephrase.
137) It is not clear what the authors mean by "the problem of AMPs in clinical application" in "While choosing safer and more selective AMPs, it also weakens the problem of AMPs in clinical application to a certain extent" (line 756)?
138) From "While choosing safer and more selective AMPs, it also weakens the problem of AMPs in clinical application to a certain extent" (line 756) is also not clear how "choosing safer and more selective AMPs" can weaken "the problem of AMPs in clinical application"? Please explain in the text.
139) Please provide introductory context for "The known mechanism of AMP membrane targeting is that cationic AMPs interact with negatively charged bacterial membranes to increase cell membrane permeability" (line 757) as it is not clear why suddenly "bacterial membranes" are being discussed.
140) Please replace "The known mechanism" with "One of the known mechanisms" (line 757).
141) Please change "that cationic AMPs interact" to "the cationic interaction of AMPs" (line 758).
142) It is not clear whether the authors are referring to outer or inner mitochondrial membrane in "The mitochondrial membrane is also a biofilm composed of a phospholipid bilayer" (line 759)?
143) Please provide reference for "The mitochondrial membrane is also a biofilm composed of a phospholipid bilayer" (line 759).
144) It is not clear whether the authors are referring to outer or inner mitochondrial membrane in "We do not know whether the mechanism of AMPs targeting to mitochondrial membrane permeability is also charge-related" (line 760).
145) "We do not know whether the mechanism of AMPs targeting to mitochondrial membrane permeability is also charge-related" (line 760) does not seem to make sense as it is hard to imagine that AMPs can be targetted to "membrane permeability".
146) Please replace "action mechanism of AMPs can we" with "the mechanism of action of AMPs, we can" (line 762).
147) Please change "them to make them" to "them" (line 763).
148) Please replace "are challenges" with "are among the challenges that" (line 764).
149) Please change "we are" to "we are currently" (line 764).
150) The statement "When MOMP is initiated by AMPs, the AMPs are stable and persistent" (line 767) is rather vague. What the authors exactly mean by stable and persistent?
151) Please replace "the AMPs" with "they" (line 768).
152) It is not exactly clear what the authors mean by "a long time" in "The pores generated directly or indirectly by Bax/Bak oligomerization exist for a long time and show a trend of continuous expansion, so MIMP is inevitable" (line 768)?
153) From "The pores generated directly or indirectly by Bax/Bak oligomerization exist for a long time and show a trend of continuous expansion, so MIMP is inevitable" (line 768) is also not clear why MIMP is inevitable following Bax/Bak oligomerization?
154) Please change "in this process, the" to "the" (line 770).
155) "affect"/"affecting" appears twice in "On the other hand, AMPs affect the opening of the MPTP by affecting the production of reactive oxygen species, the concentration of Ca2+, or targeting MPTP components" (line 771). Please fix.
156) Please replace "reactive oxygen species" with "ROS" (line 772).
157) Please change "targeting" to "targeting the" (line 773).
158) Please replace "components[73]" with "components [73]" (line 773).
159) Please change "reinforcing" to "reinforced" (line 776).
160) Please replace "the" with "these" (line 777).
161) Please change "one is apoptosis" to "apoptosis" (line 777).
162) Please replace "the other is the" with "the" (line 778).
163) Please change "apoptosis inducer (AIF)" to "AIF" (line 779).
164) Please replace "the first step of inducing apoptosis by AMPs is to directly or indirectly change the permeability of the mitochondrial membrane, thereby activating the mitochondrial pathway[77]" with "AMPs directly or indirectly change the permeability of the mitochondrial membrane, thereby activating the mitochondrial apoptotic pathway [77]" (line 781).
165) It is not clear what the authors mean by "the angle from which AMPs impact membrane permeability" in "Regardless of the angle from which AMPs impact membrane permeability, they may eventually lead to a combination of three aspects" (line 783)?
166) Please change "sometimes show that the mitochondrial structure is destroyed" to "occasionally lead to the destruction of the mitochondrial structure" (line 785).
167) It is not clear whether the authors are referring to outer or inner mitochondrial mebrane in "Therefore, AMPs that can mediate apoptosis through mitochondria, they often change the permeability of the mitochondrial membrane via a multitarget mechanism" (line 786)?
168) From "Therefore, AMPs that can mediate apoptosis through mitochondria, they often change the permeability of the mitochondrial membrane via a multitarget mechanism" (line 786) is also not clear the link to the previous sentence. Why "This may be one of the reasons why some AMPs have strong therapeutic effects and sometimes show that the mitochondrial structure is destroyed" (line 784) explains that "AMPs that can mediate apoptosis through mitochondria" often "change the permeability of the mitochondrial membrane via a multitarget mechanism"?
169) Please replace "can mediate apoptosis through mitochondria, they" with "mediate mitochondrial apoptosis" (line 786).
170) Please change "mechanism[78]" to "mechanism [78]" (line 788).
171) Please replace "action mode" with "mode of action" (line 788).
172) It is not exactly clear what the authors mean by "This" in "This facilitates the investigation of the mechanism of action of unknown AMPs and the screening and development of known AMPs" (line 790)?
173) Please change "facilitates" to "can facilitate" (line 790).
174) Please replace "apoptosis, such" with "apoptosis such" (line 792).
175) Please change "caspase3" to "caspase-3" (line 793).
176) Please replace "issues require" with "questions deserve" (line 794).
177) Please change "Caspase exist" to "Caspase present", "inducements" to "activation" 2x, "Apoptosis" to "apoptosis", "Inflammation" to "inflammation" in Figure 2.
178) Please center "AMPs" properly within the green circle in Figure 2.
179) Please move "indirect" caption so that it does not clash with one of the green arrows in Figure 2.
180) Please change "Figure 1" to "Figure 2." (line 840).
181) It is not exactly clear what the authors mean by "the other two" in "In a word, AMPs can affect one aspect and then indirectly affect the other two, causing inflammation or apoptosis in cells" (line 842)?
182) Please replace "In a word, AMPs" with "AMPs" (line 842).
183) Please change "to supporting" to "by providing the supporting" (line 845).
184) Please change "analyses, writing" to "analyses and writing up" (line 846).
185) Please replace "of" with "of the" (line 846).
186) Please change "M.L" to "M.L." (line 847).
187) Please replace "J.W., and" with "J.W. and" (line 847).
Author Response
Dear review 1:
We are grateful for your valuable suggestions and recognition. Thank you again for spending a lot of valuable time patiently reviewing the manuscript. We have solved the problem you raised one by one, all our changes of manuscript are marked up using the “Track Changes” function of MS Word. We have attached a detailed response to the document for your review.
May happiness and health be with you always.
Kind regards,
Dr. Yuzhu Han

Reviewer 2 Report
The author's response is satisfactory.
Line 127 and line 298, still show the same figure number. The author needs to pay attention.
Author Response
Dear reviewer 2
Thank you for your recognition of our work, kind comments and helpful suggestions. We carefully corrected the mistakes in the manuscript, all our changes of manuscipt are marked up using the “Track Changes” function of MS Word. With best wishes for happiness in your life and work.
With kindest regards,
Dr. Yuzhu Han
Q:Line 127 and line 298, still show the same figure number. The author needs to pay attention.
A: Line 127 is the legend of Figure 1. Line 308 (originally line 298) is the legend of Figure 3. In addition, to make it easier for the reader to understand the ways in which MPTP alters the permeability of the outer and inner mitochondrial membranes. We add a simplified diagram of the structure of MPTP in line 235 (Figure 2). Once again, we checked and corrected the errors in the manuscript.
Round 3
Reviewer 1 Report
Major points:
1) Despite the authors claim to have replaced Figure 1, the image still appears too pixelated. Provided that all suggested corrections remain the same, would it please be possible to have this figure derived from the image that appeared in the first original submission (ijms-1902926-peer-review-v1)?
2) Despite the authors claim to have replaced Figure 3 (Figure 2 in the previous round of revision), the image still appears blurry and rather pixelated. Would it please be possible to increase its pixel density and/or to redraw this figure from scratch?
Minor points:
1) Please replace "mitochondrial outer membrane" with "outer mitochondrial membrane (OMM)" (line 13).
2) Please change "mitochondrial inner membrane" to "inner mitochondrial membrane (IMM)" (line 13).
3) Please change "were" to "are" (line 15).
4) Please replace "AMPs" with "AMP" (lines 31, 35).
5) Please change "through" to "through the" (line 39).
6) From "The integrity of the mitochondrial membrane is crucial for mitochondrial apoptosis and even cell apoptosis, changes in mitochondrial membrane permeability can directly activate the mitochondrial apoptosis pathway" (line 47) is not clear whether the authors are referring to outer or inner mitochondrial membrane?
7) Please change "unknown AMPs mechanisms" to "the mechanisms of unknown AMPs" (line 63).
8) Please remove "And revealed the importance of mitochondrial integrity for apoptosis." (line 67) as it does not make sense.
9) Please replace "stimulating" with "stimulation of" or "by stimulating" (line 85).
10) Please change "during" to "during the" (line 86).
11) Please divide Table 1 into subsections entitled as "mitochondrial outer membrane", "mitochondrial inner membrane" and "mitochondrial permeability transition pore".
12) From "Inserted in the mitochondrial membrane, and changed membrane permeability" in Table 1 (MccJ25) is not clear into which mitochondrial membrane MccJ25 inserted?
13) Please replace "Adenocarcinoma A549 Cells" with "adenocarcinoma A549 cells" (17BIPHE2), "Bcl-2." with "Bcl-2" (17BIPHE2), "inner membrane structure" with "inner membrane" (Cecropin A–Melittin Hybrid Peptides), "mitochondrial outer membrane permeability" with "MOMP" (LTX-315), "U-937;" with "U-937," (BBJX), "opening." with "opening" (BIRD-2), "(HeLa);" with "(HeLa)," (Peptide B11), "mitochondria," with "mitochondria" (Scolopendin), "Cell lines" with "cell lines" (TP4), "The rat" with "Rat" (MccJ25), "membrane, and" with "membrane and" (MccJ25) in Table 1.
14) Please format "2+" in "Ca2+" using superscript in Table 1 (Scyreprocin).
15) Please define "MPTP" in the abbreviation list of Table 1.
16) Please sort abbreviation of Table 1 in an alphabetical order.
17) Please change "outer membrane of mitochondria" to "OMM" (lines 101, 112, 133).
18) Please replace "mitochondrial outer membrane" with "mitochondrial outer membrane (OMM)" (line 96) and "mitochondrial outer membrane" with "OMM" (lines 104, 106, 165, 206, 213, 301).
19) Please change "This" to "These" (line 105).
20) Please replace "that during" with "that" (line 119).
21) Please change "MIMP also" to "MIMP" (line 120).
22) There seems to be a detailed problem with the caption "(1) MOMP" in Figure 1 as "(" seems to be duplicated upon close examination. Please fix.
23) Please replace "MIMP occurs after MOMP." with "MIMP occurs after MOMP" in Figure 1.
24) Please move "tion (Figure 1) [39]." (line 131) above Figure 1.
25) Please replace "signal" with "signaling pathway" (line 137).
26) Please remove "The pathway that trigger inflammation will be covered elabrated in detail in MIMP." (line 151) as it is redundant.
27) Please replace "elabrated" with "elaborated" (line 152).
28) Please change "reaction of Cyt c with Apaf-1" to "combination of Cyt c with Apaf-1 to" (line 154).
29) Please replace "caused" with "induced" (line 155).
30) Please change "the increased" to "the" (line 155).
31) Please replace "of" with "of their" (line 157).
32) Please change "across" to "across the" (line 163).
33) Please replace "formed" with "the forming" (line 165).
34) Please change "mitochondrial inner membrane" to "mitochondrial inner membrane (IMM)" (line 166) and "mitochondrial inner membrane" to "IMM" (lines 188, 190, 195, 207).
35) Please replace "inner mitochondrial membrane" with "IMM" (line 168).
36) Please change "in" to "in mitochondrial" (line 171).
37) "evidences" could be replaced with "markers" (line 176).
38) Please replace "whether MIMP occurs" with "the occurrence of MIMP" (line 176).
39) Please change "inhibit the occurrence of" to "inhibit" (line 177).
40) Please replace "Cecropin-A–Melittin hybrid short peptide" with "Cecropin A–Melittin, a short hybrid peptide," (line 187).
41) Please change "BH3 mimicking" to "a BH3 mimetic" (line 191).
42) Please replace "MIMP on the inner mitochondrial membrane" with "MIMP" (line 203).
43) Please replace "[52- 54]" with "[52- 54]." (line 210).
44) Please change "mitochondrial permeability transition (MPT)" to "MPT" (line 211).
45) Please change "the" to "the mitochondrial" (line 212).
46) Please replace "on outer mitochondrial membrane proteins, and" with "of the OMM proteins while" (line 216).
47) Please change "accompanied," to "accompanied by the" (line 212).
48) Please replace "interfere with" with "interferes with the" (line 223).
49) Please change "damages in mitochondria" to "mitochondrial damages" (line 230).
50) Please replace "increase" with "increase in" (line 232).
51) Please change "caspase3/9" to "caspase-3/9" (lines 235, 238).
52) Please replace "OMM becomes permeable to cyt c through Bax/Bak channels" to "OMM becomes permeable to Cyt c through Bax/Bak channels" in Figure 2.
53) Please move the legend to Figure 2 just under the respective figure.
54) Please change "exhibited" to "exhibits" (line 244).
55) It is not clear what the authors mean to say by "We can infer that MIMP occurs during MPT according to the change of inner mitochondrial membrane potential" (line 247)?
56) From "We can infer that MIMP occurs during MPT according to the change of inner mitochondrial membrane potential" (line 247) is also not explicitly clear why we "can infer that MIMP occurs during MPT according to the change of inner mitochondrial membrane potential"? Please provide the missing clue.
57) Please replace "of inner" with "of" (line 248).
58) Please format the third "c" in "cytochrome c" using italics (line 253).
59) Please replace "voltage-dependent" with "Voltage-dependent" (line 253).
60) Please change "can inhibit" to "inhibits" (line 256).
61) Please change "mitochondrial outer membrane" to "OMM" (line 256).
62) From "Drugs that use the death receptor pathway to trigger the death of tumor cells and antibiotics used to treat infectious diseases can cause cells to become resistant" (line 259) is not explicitly clear to what stimulus can drugs "cause cells to become resistant"?
63) Please replace "fusion, and" with "fusion and" (line 268).
64) Please change "reduced, and" to "reduced and" (line 270).
65) From "For AMPs that can damage the mitochondrial membrane but cannot penetrate the membrane, we can use the above modification method to render them membrane permeant" (line 270) is not exactly clear what the authors refer to by "the above modification method"?
66) From "For AMPs that can damage the mitochondrial membrane but cannot penetrate the membrane, we can use the above modification method to render them membrane permeant" (line 270) is also not clear whether the authors are referring to outer or inner mitochondrial membrane? Please specify in the text.
67) Mentioning "the above modification method" does not seem to make sense in "For AMPs that can damage the mitochondrial membrane but cannot penetrate the membrane, we can use the above modification method to render them membrane permeant" (line 270) since this method is actually mentioned in the sentence below: "For example, AMP is modified by hydrophilic arginine-glycine-aspartate (RGD) sequence and has the ability to penetrate cell membrane" (line 274). Please resolve this ambiguity as it might be confusing for the reader.
68) Please replace "that can" with "that" (line 271).
69) Please change "above" to "abovementioned" (line 272).
70) Please replace "membrane" with "membranes" (line 275).
71) Please change "intracellular" to "intracellular space" (line 276).
72) Please replace "so that they can" with "to" (line 278).
73) "There are many kinds of AMPs, and the amount in the organism is small, so it is difficult to use them directly" (line 284) is puzzling as its meaning is not clear.
What do the authors mean by "kinds of AMPs", "amount in the organism", and "use them directly"? Please rephrase the sentence accordingly or remove completely to avoid confusion.
74) Please change "AMPs, and" to "AMPs and" (line 284).
75) Please replace "application" with "applications" (line 288).
76) "Linking the specific mechanism and with structures of AMPs can not only effectively study the mechanism of action of known AMPs on mitochondrial membranes but also can help to clarify the mechanism of promoting apoptosis" (line 288) does not seem to make sense as it is rather self-evident that "Linking the specific mechanism" ... can help to "study the mechanism of action" (of AMPs). Please fix.
77) Please change "effectively" to something like "effectively help to" (line 289).
78) From "Because the cytotoxicity of an AMP is positively correlated with its effect on mitochondrial membrane permeability [62], we need to select an AMP that can alter mitochondrial membrane permeability without severely damaging mitochondrial structure and without affecting other organelles" (line 290) is not clear whether the authors are referring to outer or inner mitochondrial membrane permeability?
79) From "During choosing safer and more selective AMPs problem of AMPs such as cytotoxicity and bioavailability can be weakened to a certain extent" (line 294) is not explicitly clear how "cytotoxicity and bioavailability can be weakened" (to a certain extent)?
80) Please replace "During" with "When" (line 294).
81) Please change "weakened" to "mitigated" (line 296).
82) Please replace "that" with "the" (line 297).
83) The statement that "Both cell membrane and mitochondrial outer membrane belong to biofilms" is puzzling in "Both cell membrane and mitochondrial outer membrane belong to biofilms, so it is not clear whether the interaction of AMPs on the cell membrane can also occur on the mitochondrial outer membrane" (line 298). What do the authors mean by "belong to biofilms"? Please make this unequivocally clear in the text.
84) Please change "can we" to "we can" (line 304).
85) Please replace "in" with "in mitochondrial" (line 312).
86) Please change "Ca2+, or" to "Ca2+ or" (line 314).
87) Please replace "inner and outer" with "outer and inner" (line 315).
88) Please change "the caspase-independent" to "and the caspase-independent" (line 320).
89) From "After entering cells, AMPs directly or indirectly change the permeability of the mitochondrial membrane, thereby activating the mitochondrial pathway" (line 323) is not clear whether the authors are referring to outer or inner mitochondrial membrane?
90) "Whether AMP affects MOMP, MIMP, or MPT first, these three changes may eventually occur simultaneously in mitochondria" (line 325) does not seem to make sense. Please fix.
91) "destruction" could be changed to "disruption" (line 329).
92) Please replace "mitochondria" with "mitochondrial" (line 330).
93) Please change "jointly" to "simultaneously" (line 331).
94) Please replace "action" with "action of" (line 333).
95) Please change "facilitates"to "facilitate" (line 335).
96) Please replace "lead to" with "trigger" (line 343).
97) Please change "any one of" to "either" (line 344).
98) Please replace "others" with "other pathways" (line 345).
99) Please change "X.O." to "Y.O." (line 349).
Author Response
Dear review 1:
We really appreciate your valuable advice.Thank you again for spending a lot of valuable time patiently reviewing the manuscript. We have solved the problem you raised one by one, all our changes of manuscript are marked up using the “Track Changes” function of MS Word. We have attached a detailed response to the document for your review.
May happiness and health be with you always.
Kind regards,
Dr. Yuzhu Han

Round 4
Reviewer 1 Report
Wang et al. have reviewed the molecular landscape of antimicrobial peptides (AMPs) with respect to their actions in mitochondrial membrane permeability and apoptosis. This is important since AMPs could be utilized against diseases wherein therapeutic targeting of apoptosis is beneficial such as in cancer. In their review, the authors recapitulate the basic principles of mitochondrial outer membrane permeability (MOMP), mitochondrial inner membrane permeability (MIMP), and mitochondrial permeability transition (MPT) and how these pathways converge on caspases to elicit apoptosis. Intriguingly, they suggest that MOMP, MIMP, and MPT could be triggered by AMPs simultaneously. Although such synergistic effect could be potentially exploited for improved chemotherapy, more studies are required to gain a more comprehensive understanding of the underlying mechanism of action of AMPs. In conclusion, Wang et al. have sparked a renewed discussion on mitochondrial membranes as the epicenter for targeted therapy and, as such, their manuscript has the profound capacity to attract further interest of into this exciting and rather underappreciated field.
1) Please replace "mitochondria-induced" with "mitochondria-mediated" (line 9).
2) Please change "even" to "even on" (line 26).
3) Please replace "producing or reducing" with "altering" (line 27).
4) Please change "but" to "but can" (line 29).
5) Please replace "diseases by the intracellular pathway" with "diseases" (line 41).
6) Please replace "and" with "and to" (line 42).
7) Please change "apoptosis, changes" to "apoptosis. Changes" (line 47).
8) Please replace "pores" with "pore" (line 53).
9) Please change "and" to "and the" (line 73).
10) Please replace "AMPs can" with "AMPs" (line 75).
11) Please change "and" to "or" (line 79).
12) Please replace "outer mitochondrial membrane" with "outer mitochondrial membrane (OMM)" (line 81) and "mitochondrial outer membrane (OMM)" with "OMM" (line 92).
13) Please replace "stimulation of" with "by stimulating" (line 82).
14) Please change "mitochondrial" to "mitochondrial apoptotic" (line 83).
15) Please replace "leukemia, CCRF-CEM" with "leukemia CCRF-CEM" (PETK), "A–Melittin" with "A-melittin" (Cecropin A–Melittin Hybrid Peptides), "BIRD-2 provokes" with "Provoked" (BIRD-2), "overload, followed" with "overload followed" (BIRD-2), "Bax, and" with "Bax and" (Iturin), "produced, followed" with "produced followed" (Mt6-21DLeuJ), "homeostasis" with "homeostasis," (Scolopendin), "polarization," with "polarization and" (Surfactin), "is produced, followed" with "was produced followed" in Table 1.
16) Please format "c" in "Cyt c" using italics in Table 1 (LL-37, LTX-315, BMAP-28, Iturin, Scolopendin, Surfactin).
17) Please format "c" in "Cyt c" using italics (lines 88, 106, 107, 123, 124, 130, 138, 150, 233, 243, 303).
18) Please format the third "c" in "cytochrome c" using italics (lines 88, 91, 106).
19) Please change "(cytochrome c and apoptosis-inducing factor) existing in the mitochondrial membrane gap" to "existing in the mitochondrial membrane gap (cytochrome c and apoptosis-inducing factor)" (line 91).
20) Please replace "mitochondrial membrane gap" with "intermembrane space (IMS)" (line 92) and "intermembrane space (IMS)" with "IMS" (line 158).
21) Please change "inner membrane" to "inner mitochondrial membrane (IMM)" (line 95), "inner membrane of mitochondria" to "IMM" (line 108), "inner membrane" to "IMM" (lines 110, 187, 192), and "mitochondrial inner membrane (IMM)" to "IMM" (line 161).
22) Please replace "Bax can" with "Bax" (line 98).
23) Please change "outer membrane of mitochondria" to "OMM" (line 98).
24) Please replace "Bcl-LX" with "Bcl-XL" (line 99).
25) Please change "that can" to "that" (line 99).
26) Please replace "in the same oligomerization manner" with "by the same oligomerization mechanism" (line 102).
27) Please change "induced" to "mediated" (line 103).
28) Please replace "These artificially modified AMPs targeting the OMM are good candidate drugs for the treatment of tumors and infectious diseases. LL-37 can accumulate in the mitochondria of human osteogenic MG63 cells and promote the release of cytochrome c (Cyt c) and apoptosis-inducing factor (AIF) between the outer and inner mitochondrial membranes, while Cyt c oxidase IV (COXIV) located in the inner membrane of mitochondria was not detected. This finding indicates that LL-37 may only activate the permeability of the OMM without damaging the inner membrane structure [34]." with "LL-37 can accumulate in the mitochondria of human osteogenic MG63 cells and promote the release of cytochrome c (Cyt c) and apoptosis-inducing factor (AIF) between the outer and inner mitochondrial membranes, while Cyt c oxidase IV (COXIV) located in the inner membrane of mitochondria was not detected. This finding indicates that LL-37 may only activate the permeability of the OMM without damaging the inner membrane structure [34]. These artificially modified AMPs targeting the OMM are good candidate drugs for the treatment of tumors and infectious diseases." (line 103).
29) Please provide reference for "LL-37 can accumulate in the mitochondria of human osteogenic MG63 cells and promote the release of cytochrome c (Cyt c) and apoptosis-inducing factor (AIF) between the outer and inner mitochondrial membranes, while Cyt c oxidase IV (COXIV) located in the inner membrane of mitochondria was not detected" (line 105).
30) "LL-37 can accumulate in the mitochondria of human osteogenic MG63 cells and promote the release of cytochrome c (Cyt c) and apoptosis-inducing factor (AIF) between the outer and inner mitochondrial membranes, while Cyt c oxidase IV (COXIV) located in the inner membrane of mitochondria was not detected" (line 105) does not seem to make sense for the two following reasons:
a) Cyt c and AIF are residents of the mitochondrial intermembrane space under normal conditions, therefore it is hard to conceive that they can be released "between the outer and inner mitochondrial membranes" as they are already present there.
b) "Cyt c oxidase IV (COXIV) located in the inner membrane of mitochondria was not detected" does not fit the sentence since the explanatory context of why and where COXIV was not detected is lacking. In fact, do the authors mean to say that Cyt c and AIF were detected? Again, this has to be specified including the respective compartments, in which these proteins were detected.
31) Please change "can accumulate" to "accumulates" (line 105).
32) Please replace "promote" with "promotes" (line 106).
33) Please change "Cyt c" to "cytochrome c" (line 107).
34) Please replace "of apoptosis-related proteins" with "of" (line 113).
35) Please change "simultaneously occur in mitochondria" to "occur simultaneously" (line 118).
36) Please move "tivate the caspase cascade and lead to apoptosis [38]. At the same time, Smac can prevent the anti-apoptotic effect of XIAP. If caspase is absent or inactivated during MOMP, several apoptotic factors can induce apoptosis via inflammation (Figure 1) [39]." (line 125) underneath the legend of Figure 1.
37) Please replace "Cyt C" with "Cyt c" (line 129).
38) Please format the second "C" in "Cyt C" using italics (line 129).
39) Please change "signal" to "signaling" (line 133).
40) Please replace "pathway" with "manner" (line 139).
41) Please change "action mechanism" to "mechanism of action" (line 140).
42) Please replace "17BIPHE2," with "17BIPHE2 and" (line 140).
43) Please change "AMPs can" to "AMPs" (line 141).
44) Please replace "two" with "two different" or "two distinct" (line 141).
45) Please change "to directly participate" to "direct participation of AMPs" (line 142).
46) Please replace "to" with "to trigger" (line 150).
47) Please change "existence" to "simultaneous existence" (line 153).
48) Please replace "and" with "and during which" (line 157).
49) Please change "outer membrane" to "OMM" (line 162).
50) Please replace "which is transmitted" with "the signal of which is transmitted" or "which signals" (line 167).
51) Please change "A–Melittin" to "A-melittin" (line 181).
52) Please replace "peptide" with "peptide," (line 181).
53) Please change "substances" to "factors" (line 183).
54) It is not clear what the authors mean by "BH3" in "In a study of mitochondria-mediated apoptosis, when cells were treated with BH3 mimicking mimetic inhibitor ABT-737, we observed that MOMP was followed by mtDNA leakage" (line 184) since this was nowhere explained?
55) Please replace "with" with "with the" (line 185).
56) Please change "can maintain" to "maintains" (line 212).
57) Please replace "the with" with "the" (line 215).
58) Please change "causes" to "induces" (line 219).
59) Please replace "MPTP, and" with "MPTP and" (line 222).
60) Please change "indicated" to "indicate" (line 231).
61) Please replace "the" with "is their" (line 233).
62) Please change "increased, and" to "increased and" (line 234).
63) Please move "and MIMP, besides inducing apoptosis in the same manner. In addition, AMPs strengthened the necessary link between MPTP and MOMP (Figure 1). We can infer that MIMP 237 occurs during MPT. It is not clear whether MIMP is directly caused by the continuous opening of MPTP or by MOMP." (line 236) underneath the legend of Figure 2.
64) Please replace "use" with "exploit" (line 249).
65) Please change "the membrane" to "it" (line 259).
66) Please replace "AMP is modified by hydrophilic arginine-glycine-aspartate (RGD) sequence by protein fusion and has" with "AMPs modified by hydrophilic arginine-glycine-aspartate (RGD) sequence by protein fusion have" (line 261).
67) Please change "delivers AMP" to something like "allows for the transport of AMPs" or "facilitates the transport of AMPs" (line 262).
68) "It is also possible to reasonably design AMPs to render them membrane permeant or to enhance their binding to mitochondrial targets [70], which is of great significance for improving their therapeutic effect" is not exactly precise since the fact that it is "possible to reasonably design AMPs to render them membrane permeant" was already indicated in the previous text: "For example, AMP is modified by hydrophilic arginine-glycine-aspartate (RGD) sequence by protein fusion and has the ability to penetrate cell membranes. This modification delivers AMP from the extracellular to the intracellular space [69]" (line 260). In this context, the sentence therefore does not seem to make sense. Please fix.
69) Please replace "according to Table 1, almost all AMPs that can change mitochondrial membrane permeability have α-helices or can form ring structures and their targets may be related to the structure and activity of AMPs" with "almost all AMPs that can change mitochondrial membrane permeability have α-helices or can form ring structures and their targets may be related to the structure and activity of AMPs (Table 1)" (line 267).
70) Please change "choose" to "therefore select" (line 279).
71) Please replace "increase membrane" with "increase their" (line 283).
72) Please change "charges" to "charge" (line 285).
73) Please replace "the" with "similar" (line 285).
74) Please change "AMPs on" to "AMPs with" (line 286).
75) Please replace "can conclude that AMPs are" with "conclude that AMPs are potent" (line 292).
76) It is not clear what the authors are referring to as "pores generated ... indirectly by Bax/Bak oligomerization" in "We can conclude that AMPs are inducers of apoptosis. If AMPs are stable and persistent to induce MOMP, the pores generated directly or indirectly by Bax/Bak oligomerization exist for a long time and show a trend of continuous expansion until AMPs is inactive, so MIMP is inevitable" (line 292)?
77) Please change "Bax/Bak" to "Bax or Bak" (line 293).
78) Please replace "[75]; And" with "[75] and" (line 303).
79) Please replace "Whether AMP affects" with "Irrespectively of whether AMPs affect" (line 308).
80) Please change "manner of multi-path interaction" to "synergistic manner" (line 313).
81) It is not exactly clear what the authors mean by "the increase in caspase-3" in "Undeniably, there are many unrevealed relationships in the effect of AMPs on apoptosis such as whether the increase in caspase-3 is directly related to AMPs and whether AMPs are directly related to MIMP" (line 317)? Are they referring to increased caspase-3 expression or activity?
82) Please replace "of the others pathways" with "processes" (line 324).
Author Response
Dear reviewer 1:
We are grateful for your valuable suggestions and recognition. Thank you again for spending a lot of valuable time patiently reviewing the manuscript. We have solved the problem you raised one by one, all our changes of manuscript are marked up using the “Track Changes” function of MS Word. Clear figures are in the word file. We have attached a detailed reply to the document for your review.
May happiness and health be with you always.
Kind regards,
Dr. Yuzhu Han
